# A PICTURE OF THE SPACE OF TYPICAL LEARNING TASKS

## ABSTRACT

We develop a technique to analyze representations learned by deep networks when they are trained on different tasks using supervised, meta- and contrastive learning. We develop a technique to visualize such representations using an isometric embedding of the space of probabilistic models into a lower-dimensional space, i.e., one that preserves pairwise distances. We discover the following surprising phenomena that shed light upon the structure in the space of learning tasks: (1) the manifold of probabilistic models trained on different tasks using different representation learning methods is effectively low-dimensional; (2) supervised learning on one task results in a surprising amount of progress on seemingly dissimilar tasks; progress on other tasks is larger if the training task has diverse classes; (3) the structure of the space of tasks indicated by our analysis is consistent with parts of the Wordnet phylogenetic tree; (4) fine-tuning a model upon a sub-task does not change the representation much if the model was trained for a large number of epochs; (5) episodic meta-learning algorithms fit similar models eventually as that of supervised learning, even if the two traverse different trajectories during training; (6) contrastive learning methods trained on different datasets learn similar representations. We use classification tasks constructed from the CIFAR-10 and Imagenet datasets to study these phenomena.

## 1 INTRODUCTION

Exploiting data from related tasks to reduce the sample complexity of learning a desired task, is an idea that lies at the heart of burgeoning fields like transfer, multi-task, meta, few-shot, and self-supervised learning. These algorithms have shown an impressive ability to learn representations that can predict well on new tasks. The algorithms are very diverse in how they work but it stands to reason they must be exploiting some shared structure in the space of learning tasks. Although there is a large body of work that seeks to understand relatedness among tasks and how these algorithms exploit it (see §4 for a discussion of related work), we do not know what this shared structure precisely is. Our work makes the following contributions to advancing this line of research.

**We develop a technique to analyze the learned representation on a task, and its relationship to other tasks.** Our key technical innovation is to use ideas from information geometry to characterize the geometry of the space of probabilistic models fit on different tasks. We develop methods to embed training trajectories of probabilistic models into a lower-dimensional space isometrically, i.e., while preserving pairwise distances. This allows us to faithfully visualize the geometry of these very high dimensional spaces (for Imagenet, our probabilistic models are in $\sim 10^7$ dimensions) and thereby interpret the geometry of the space of learning tasks. These technical tools are very general and shed light on the shared structure among tasks.

**We point these technical tools to study how algorithms that learn from multiple tasks work. We provide evidence for the following phenomena.**

(1) The manifold of probabilistic models trained on different tasks using different representation learning methods is effectively low-dimensional, and this dimensionality is rather small; For Imagenet, a 3-dimensional subspace preserves 80.02% of the pairwise distances between models, which we define (in Appendix D) as the "explained stress";

(2) Supervised learning on one task results in a surprising amount of progress (informally, "progress" means that the representation learned on one can be used to make accurate predictions on other tasks; this is defined precisely in (4)) on seemingly dissimilar tasks; progress on other tasks is larger if the training task has diverse classes;

(3) The structure of the space of tasks indicated by our analysis is consistent with parts of the Wordnet phylogenetic tree;

(4) Fine-tuning a model upon a sub-task does not change the representation much if the model was trained for a large number of epochs;

(5) Episodic meta-learning algorithms fit similar models eventually as that of supervised learning, even if the two traverse different trajectories during training;

(6) Contrastive learning methods trained on different datasets learn similar representations.

We demonstrate these findings on image classification tasks constructed from CIFAR-10 and Imagenet datasets.

## 2 METHODS

**Modeling the task**  We define a task $P$ as a joint distribution on inputs $x \in \mathbb{R}^d$ and outputs $y \in \{1, \ldots, C\}$ corresponding to $C$ classes. Suppose we have $N$ independent and identically distributed samples $\{(x_n, y_n^*)\}_{n=1}^N$ from $P$. Let $\vec{y} = (y_1, \ldots, y_N)$ denote any sequence of outputs on these $N$ samples and $\vec{y^*}$ denote the sequence of ground-truth labels. We may now model the task as

$$P_w(\vec{y}) = \prod_{n=1}^N p_w^n(y_n) \tag{1}$$

where $w$ are the parameters of the model and we have used the shorthand $p_w^n(y_n) \equiv p_w(y_n \mid x_n)$. The true probability distribution which corresponds to the ground-truth labels is denoted by $P_* \equiv P(\vec{y^*})$. In the same way, let us denote by $P_0$ the probability distribution that corresponds to $p^n(y) = 1/C$ for all $n$ and all $y$, i.e., $P_0$ predicts accurately on a fraction $1/C$ of the samples.

**Bhattacharyya distance**  Given two models $P_u$ and $P_v$ parameterized by weights $u$ and $v$ respectively, we define the Bhattacharyya distance (Bhattacharyya, 1946) between them averaged over samples as

$$\begin{aligned} d_B(P_u, P_v) &:= -N^{-1} \log \sum_{\vec{y}} \prod_n \sqrt{p_u(y_n)\, p_v(y_n)} \\ &\overset{(*)}{=} -N^{-1} \log \prod_{n=1}^N \sum_{c=1}^C \sqrt{p_u^n(c)\, p_v^n(c)} \\ &= -N^{-1} \sum_n \log \sum_c \sqrt{p_u^n(c)\, p_v^n(c)}; \end{aligned} \tag{2}$$

see Appendix C for more details on $(*)$. Our model (1) involves a product over the probabilities of $N$ samples. Typical distances for probability distributions, e.g., Hellinger distance $2\left(1 - \prod_n \sum_c \sqrt{p_u^n(c)\, p_v^n(c)}\right)$, saturate when the number of  samples $N$ is large (because random high-dimensional vectors are nearly orthogonal). It is thus difficult to use such distances to understand high-dimensional probabilistic models. However, the Bhattacharyya distance is well-behaved for large  $N$ due to the logarithm (Quinn et al., 2019; Teoh et al., 2020), and that is why it is well suited to our problem.

**Distances between trajectories of probabilistic models**  Consider a trajectory $(w(k))_{k=0,\ldots,T}$ that records the weights after $T$ updates of the optimization algorithm, e.g., stochastic gradient descent. This trajectory corresponds to a trajectory of probabilistic models $\tilde{\tau}_w = (P_{w(k)})_{k=0,\ldots,T}$. We are interested in calculating distances between such training trajectories. First, consider $\tilde{\tau}_u = (u(0), u(1), u(2), \ldots, u(T))$ and another trajectory $\tilde{\tau}_v \equiv (u(0), u(2), u(4), \ldots, u(T), u(T), \ldots, u(T))$ which trains twice as fast but to the same end point. If we define the distance between these trajectories as, say, $\sum_k d_B(P_{u(k)}, P_{v(k)})$, then the distance between $\tilde{\tau}_u$ and $\tilde{\tau}_v$ will be non-zero—even if they are fundamentally the same. This issue is more pronounced when we calculate distances between training trajectories of different tasks. It arises because we are recording each trajectory using a different time coordinate, namely its own training progress.

To compare two trajectories correctly, we need a notion of time that can allow us to uniquely index any trajectory. The geodesic between the start point $P_0$ and the true distribution $P_*$ is a natural candidate for this purpose. Geodesics are locally length-minimizing curves in a metric space. For the product manifold in (1), we can obtain a closed-form formula for the geodesic as follows. We can think of the square root of the probabilities $\sqrt{p_u^n(c)}$ as a point on a $C$-dimensional sphere. Given two models $P_u$ and $P_v$ the geodesic connecting them under the Fisher information metric is the great circle on this sphere (Ito & Dechant, 2020, Eq. 47):

$$\sqrt{P_{u,v}^\lambda} = \prod_n \left( \frac{\sin\left((1-\lambda)d_G^n\right)}{\sin\left(d_G^n\right)} \sqrt{p_u^n} + \frac{\sin\left(\lambda d_G^n\right)}{\sin\left(d_G^n\right)} \sqrt{p_v^n} \right); \text{ for } \lambda \in [0, 1], \tag{3}$$

where $d_G = \sum_n \cos^{-1}\left(\sum_c \sqrt{p_u^n(c)}\sqrt{p_v^n(c)}\right)$ is one half of the great-circle distance on the $N$-way product manifold of probability distributions $p_u^n(\cdot)$ and $p_v^n(\cdot)$.

**Every point $w$ on the trajectory $\tilde{\tau}_w$ can now be reindexed by a new time which we call "progress"**

$$t_w = \inf_{\lambda \in [0,1]} d_G(P_w, P_{0,*}^\lambda). \tag{4}$$

Note that the progress $t_w \in [0,1]$ for any point on any trajectory. In practice, we solve (4) using bisection search (Brent, 1971). Observe that (3) also allows us to calculate the geodesic between two successive points $P_{w(k)}$ and $P_{w(k+1)}$ of the trajectory. Any point on this geodesic $P_{w(k),w(k+1)}^\lambda$ for $\lambda \in [0,1]$ can be assigned a progress $t_{w(k)} + \lambda(t_{w(k+1)} - t_{w(k)})$. We have effectively converted our trajectory which is a discrete sequence of models $\tilde{\tau}_w = (P_{w(k)})_{k=0,\dots,T}$ into a continuous curve $\tau_w = (P_{w(t)})_{t \in [0,1]}$. We can now calculate the distance between trajectories $\tau_u$ and $\tau_v$ as

$$d_{\text{traj}}(\tau_u, \tau_v) = \int_0^1 d_B(P_{u(t)}, P_{v(t)})\, dt\,; \tag{5}$$

we approximate this integral in practice using a uniform grid on $[0,1]$.

**Riemann length of a trajectory of probabilistic models**  Divergences like the Bhattacharyya distance or the Kullback-Leibler (KL) distance (which is exactly the cross-entropy loss)

$$d_{\text{KL}}(P_*, P_w) = -N^{-1}\sum_n \sum_c p_*^n(c) \log p_w^n(c) - H(P_*);$$

where $H(P_*)$ is the entropy of $P_*$ (and independent of $w$) can be used to define a Riemannian structure in the space of probabilistic models (Amari, 2016). The distance between two models $P_w$ and $P_{w+dw}$ parametrized by infinitesimally different weights is

$$ds^2 = 4d_B(P_w, P_{w+dw}) = \langle dw\,, g(w)\, dw\rangle + \mathcal{O}(\|dw\|^2),$$

where $g(w) = N^{-1}\sum_{\vec{y}}(P_w)^{-1}\partial^2 P_w$ is the Fisher Information Matrix (FIM). This FIM is therefore the metric of the space of the probability distributions and weights $w$ play the role of the coordinates in this space. The Bhattacharyya distance and the KL-divergence induce the same metric up to a scalar factor. We can therefore calculate the Riemann length of a trajectory $\tau_w$ by integrating the infinitesimal lengths

$$\text{Length}(\tau_w) = 2\int_0^1 \sqrt{d_B(P_{w(t)}, P_{w(t+dt)})}. \tag{6}$$

Observe that we do not need the FIM to calculate the length. We can think of the length of a trajectory taken by a model to reach the solution $P_*$ compared to the length of the geodesic as a measure of the inefficiency of the training procedure. This inefficiency can arise because: (a) not all probability distributions along the geodesic can be parametrized by our model class (approximation error), and (b) the training process may take steps that are misaligned with the geodesic (e.g., due to the loss function, mini-batch updates, supervised vs. some other form of representation learning, etc.).

**Mapping a model trained on one task to another task using "imprinting"**  We will consider different tasks $\{P^k\}_{k=1,\dots,}$ with the same input domain but possibly different number of classes $C^k$. Given a model $P_w^1$ parametrized by weights $w$ for one task, we are interested in evaluating its learned representation on another task, say, $P^2$. There are many ways of doing so, e.g., one could fine-tune the weights using data from $P^2$. We use a simple technique that re-initializes the final layer of the model. Let us separate the weights into two parts $w = (w_1, w_2)$ for the backbone and the classifier respectively. Let $\varphi(x; w_1)$ denote the features of the penultimate layer corresponding to an input $x$. To clarify, the logits are given by $\mathbb{R}^{C^1} \ni w_2^\top \varphi(x; w_1)$ (with perhaps an added bias) and the output $p_w(c \mid x_n)$ for $c = 1,\dots,C^1$ is computed using a softmax applied to these logits.

If we have learned $w$ from one task $P^1$, then we can re-initialize each row of the classifier weights $(w_2)_c'$ for $c = 1,\dots,C^2$ to maximize the cosine similarity with the average feature of samples from task $P^2$ with ground-truth class $c$:

$$(w_2)_c' = h/\|h\|_2 \quad \text{where } h = \sum_{\{x : y_x^* = c\}} \varphi(x; w_1). \tag{7}$$

The new model $w = (w_1, w_2')$ can now be used to make predictions on the task $P^2$. This technique for mapping the learned representation from one task to another is motivated from previous work (Dhillon et al., 2020; Qi et al., 2018; Hu et al., 2015) which initializes the final layer of a model before adapting it to new classes using fine-tuning. It is called imprinting because rows of the classifier's weights can be thought of as templates of different classes against which features are compared.

Using imprinting, we can map a trajectory $\tau_w^1$ of a model being trained on task $P^1$ to another task $P^2$ by mapping each point along the trajectory; let us denote this mapped trajectory by $\tau_w^{1 \to 2}$.

**Remark 1 (Imprinting vs. training the final layer).** We are interested in mapping 1000s of models across different trajectories to other tasks. Training the final layer for all these models is cumbersome and we can avoid doing so using imprinting. Note that if we were to train the classifier $w_2$ (with backbone $w_1$ fixed) using samples from the other task under the constraint that rows of $w_2$ have unit $\ell_2$ norm, then we would obtain the imprinted weights as our solution (see Appendix E).

**How to choose an appropriate task to map different models to?** Consider the training trajectory $\tau_u^1$ of a model trained on task $P^1$ and another trajectory $\tau_v^2$ of a model trained on task $P^2$. Using the procedure in (7), we can map these trajectories to the other task to get $\tau_u^{1 \to 2}$ and $\tau_v^{2 \to 1}$. This allows us to calculate $d_{\text{traj}}(\tau_u^{1 \to 2}, \tau_v^2)$ which is the distance of the trajectory of the model trained on $P^1$ and then mapped to $P^2$ with respect to the trajectory of a model trained on task $P^2$; we can calculate $d_{\text{traj}}(\tau_v^{2 \to 1}, \tau_u^1)$ analogously. Roughly speaking, if the two learning tasks $P^1$ and $P^2$ are very different, (e.g., Animals in CIFAR-10 and Vehicles in CIFAR-10), then these distances will be large.

The distances $d_{\text{traj}}(\tau_u^{1 \to 2}, \tau_v^2)$ and $d_{\text{traj}}(\tau_v^{2 \to 1}, \tau_u^1)$ are reasonable candidates to study similarities between tasks $P^1$ and $P^2$, but they are not equal to one another. We are also interested in doing such calculations with models trained on many different tasks, and mapping them to each other will lead to an explosion of quantities. To circumvent this, we map to a unique task whose output space is the union of the output spaces of the individual tasks, e.g., to study $P^1$ (Animals) and $P^2$ (Vehicles), we will map both trajectories to $P^U$ which is all of CIFAR-10. We calculate quantities like

$$d_{\text{traj}}(\tau_u^{1 \to U}, \tau_v^{2 \to U}). \tag{8}$$

**Embedding a probabilistic model in lower-dimensions** We use a technique called intensive principal component analysis (InPCA) (Quinn et al., 2019) to embed a probabilistic model into a lower-dimensional space. This enables us to visually inspect relationships between different tasks and representation learning methods. For $m$ probability distributions, consider a matrix $D \in \mathbb{R}^{m \times m}$ with entries $D_{uv} = d_{\text{B}}(P_u, P_v)$ and

$$W = -LDL/2 \tag{9}$$

where $L_{ij} = \delta_{ij} - 1/m$ is the centering matrix. An eigen-decomposition of $W = U\Sigma U^\top$ where the eigenvalues are sorted in descending order of their magnitudes $|\Sigma_{00}| \geq |\Sigma_{11}| \geq \ldots$ allows us to compute the embedding of these $m$ probability distributions into an $m$-dimensional space as $\mathbb{R}^{m \times m} \ni X = U\sqrt{\Sigma}$. Unlike standard PCA where eigenvalues are non-negative, eigenvalues of InPCA can be both positive and negative. There are technical reasons for this (Quinn et al., 2019). But such an embedding into a Minkowski space allows InPCA to be an isometry:

$$\sum_i (X_u^i - X_v^i)^2 = d_{\text{B}}(P_u, P_v) \geq 0 \tag{10}$$

for embeddings $X_u, X_v$ of any two probability distributions $P_u$ and $P_v$. This property of InPCA is akin to that standard PCA, where if we preserve all the eigenvectors (i.e., $k = n$), then (10) holds, but if we use fewer eigenvectors to project the points, then pairwise distances can be distorted. We only use InPCA for visualization, all our findings are quantitative and do not rely on embedding into lower dimensions. We quantify how well pairwise distances are preserved by a $k$-dimensional InPCA embedding using a quantity called "explained stress" ($\chi_k$):

$$\chi_k = 1 - \frac{\|W - \sum_{i=1}^k \Sigma_{ii} \, U_i U_i^\top\|_{\text{F}}}{\|W\|_{\text{F}}} = 1 - \sqrt{\frac{\sum_{i=k+1}^m \Sigma_{ii}^2}{\sum_{i=1}^m \Sigma_{ii}^2}}.$$

Appendix A.2 describes how we implement InPCA for very high-dimensional probabilistic models.

## 3 Results

We next discuss our key results. Our experiments were conducted using multiple neural architectures and image-classification tasks created from the CIFAR-10 and Imagenet datasets; see Appendix A.

All the analysis in this paper (except Fig. 2) was conducted using the test data. All models were trained using the training data, but all mapped models, distances between trajectories, quantitative evaluation of progress and InPCA embeddings were computed using the test dataset. The reason for this is that we would like to study the geometry of tasks as evidenced by samples that were not a part of training. To emphasize, we do not develop any new algorithms for learning in this paper and therefore using the test data to quantify relationships between tasks is reasonable; see similar motivations in Kaplun et al. (2022) among others. Our findings remain valid when training data is used for analysis; this is primarily because in most of our experiments, a representation is trained on one task but makes predictions on a completely new task after mapping.

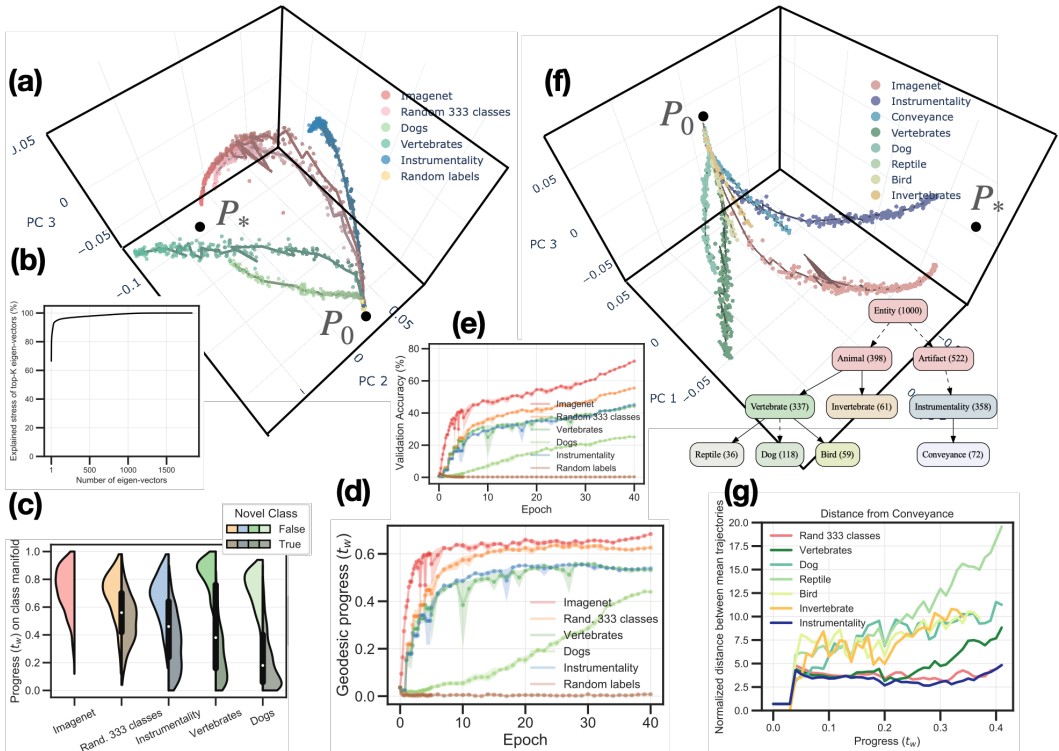

**Figure 1: (a)** InPCA embedding of training trajectories of probabilistic models trained on 6 tasks from Imagenet. Each point is one network, bold lines connect the averages of probabilities of 5 training trajectories of each task (5 random initializations in weight space). The point $P_0$ denotes start of training (complete ignorance) and $P_*$ is drawn to show the model that predicts perfectly (truth). All trajectories move towards the end point of training on the entire Imagenet. Training on one task makes a remarkable amount of progress on unseen, seemingly dissimilar, classes. Trajectories of models trained on a random set of 333 classes are similar to those of the entire Imagenet. Some classes (Instrumentality) are closer to this trajectory while others such as Vertebrates and Dogs are farther away. Dogs is a semantic subset of Vertebrates; it splits at the beginning but seems to eventually reach a similar representation as one of the intermediate points of Vertebrates. See Result 1.

**(b)** Percentage explained stress (defined in (12)) captured by subspace spanned by the top $k$ InPCA eigenvectors. Explained stress measures if the Bhattacharyya distances between models are accurately captured in this sub-space. Top 3 dimensions capture more than 80% of the explained stress, and thus the manifold is effectively low-dimensional. See Result 1.

**(c)** If we train a model on Vertebrates, do we make progress on unseen classes? This plot measures the progress made by each model on classes seen during training (left half, lighter shade) and on novel classes (right half, darker shade). We compute $t_w^c$ which is the progress $t_w$ of images restricted to a single class $c$. This quantity $t_w^c$ measures the quality of the representation for class $c$. The violin plots that denote the distribution of $t_w^c$ indicate that we make more progress on classes seen during training. If the model sees a larger diversity of classes (like with random 333 classes), more progress is made on the novel classes. Surprisingly, even if we train on just the "Dogs", we make some progress on novel classes.

**(d)** Progress $t_w$ (4) on the Y-axis against the number of epochs of training on the X-axis. The progress $t_w$ increases with more epochs of training—all models make non-trivial progress towards the truth $P_*$ ($t_w = 1$). Surprisingly, even if we train on only Dogs (118 classes) we make progress on the entire Imagenet. **(e)** . Validation accuracy as a function of epochs. See Result 2.

**(f)** Trajectories of models trained on different phyla of Wordnet (colors match those of the inset graph). The model manifold is again effectively low-dimensional (78.72% explained stress in 3 dimensions). Trajectories of tasks that are nearby in Wordnet are also nearby, and vice-versa. **(g)** Bhattacharya distance between the mean trajectories (over random initializations) on different tasks and the mean trajectory of Conveyance. This distance is normalized by the average of the tube radii (maximum distance of one of the 5 trajectories from the mean, computed at each progress) of the two trajectories. Such quantities allow us to make precise statements about the differences between representations and show some very surprising conclusions. E.g., trajectories of random 333 classes (light orange) are closer to Conveyance (as expected), but those of Vertebrates (red) are equally far away for more than 60% ($t_w \approx 0.25$) of the progress. In other words, training on Vertebrates (reptiles, dog, bird) makes a remarkable progress on Conveyance (cars, planes). See Result 2.

**Result 1. The manifold of models trained on different tasks, and using different representation learning methods, is effectively low-dimensional.** We trained multiple models on 6 different sub-tasks of Imagenet (from 5 random initializations each) to study the dimensionality of the manifold of probabilistic models along the training trajectories (100 points equidistant in progress (4)) after mapping all models to all Imagenet classes ($\sim 10^8$ dimensions). We use the explained stress (defined in Appendix D), to measure if the distances are preserved by the first $k$ dimensions of the embedding of the models. The first 3 dimensions of InPCA (Fig. 1a) preserve 80% of the explained stress (Fig. 1b shows more dimensions). This is therefore a remarkably low-dimensional manifold. It is not exactly low-dimensional because the explained stress is not 100%, but it is an effectively low-dimensional manifold. This experiment also indicates that the individual manifolds of models trained on one task are low-dimensional, even if they start from different random initializations in the weight space. Such low-dimensional manifolds are seen in *all* our experiments, irrespective of the specific method used for representation learning, namely, supervised, transfer (fine-tuning), meta and contrastive learning.

**Result 2. Supervised learning on one task results in a surprising amount of progress on seemingly dissimilar tasks. Progress on other tasks is larger if the training task has diverse classes.** We studied the progress $t_w$ (4) made by models (Fig. 1d) trained on tasks from Result 1. Training on the task "Dogs" makes progress on other tasks, even seemingly dissimilar ones like "Instruments" (which contains vehicles, devices and clothing). In fact, it makes progress on the entire Imagenet (about 63.38% of the progress of a model trained directly on Imagenet). Progress is larger for larger phyla of Imagenet (Vertebrates and Instruments). But what is surprising is that if we train on a random subset of 333 classes (a third of Imagenet), then the progress on the entire Imagenet is very large (92%). This points to a remarkably strong shared structure among classes even for large datasets such as Imagenet. Note that this *does not* mean that tasks such as Vertebrates and Instruments are similar to each other. Even if training trajectories are similar for a while, they do bifurcate eventually and the final models are different (Fig. 1g).

To study this further, we projected models trained on one task onto the geodesics of unseen classes (Fig. 1c) calculated using (3)). A model trained on the entire Imagenet makes uneven progress on the various classes (but about 80% progress across them, progress is highly correlated with test error of different classes). Models trained on the 6 individual tasks also make progress on other unseen classes. As before, training on Instruments, Vertebrates, Dogs makes smaller progress on unseen classes compared to training on a random subset of 333 classes. This is geometric evidence that the more diverse the training dataset, the better the generalization to *unseen* classes/tasks; this phenomenon has been widely noticed and utilized to train models on multiple tasks as we discuss in §4.

**Result 3. The structure of the space of tasks indicated by our visualization technique is consistent with parts of the Wordnet phylogenetic tree** To obtain a more fine-grained characterization of how the geometry in the space of learnable tasks reflects the semantics of these tasks, we selected two particular phyla of Imagenet (Animals, Artifacts) and created sub-tasks using classes that belong to these phyla (Fig. 1f). Trajectories of models trained on Instruments and Conveyance are closer together than those of Animals. Within the Animals phylum, trajectories of Vertebrates (Dog, Reptile, Bird) are closer together than those of Invertebrates (Fig. 1g for quantitative metrics). Effectively, we can recover a part of the phylogenetic tree of Wordnet using our training trajectories. We speculate that this may point to some shared structure between visual features of images and natural language-based semantics of the corresponding categories which was used to create Wordnet (Miller, 1998) of the corresponding categories. This alignment with a natural notion of relatedness also demonstrates the effectiveness of our methods to understand the structure in the space of visual tasks.

**Result 4. Fine-tuning a pre-trained model on a sub-task does not change the representation much.** To understand how models train on multiple tasks, we selected two binary classification sub-tasks of CIFAR-10 (Airplane vs. Automobile, and Bird vs. Cat). We selected models at different stages of standard supervised learning on CIFAR-10 (i.e., using 10-way output and softmax cross-entropy loss) and fine-tuned each of these models on two sub-tasks (the entire network is fine-tuned without freezing the backbone). As Fig. 3 shows, models that were fine-tuned from earlier parts of the trajectory travel a large distance and move away from trajectories of the supervised learned CIFAR-10 models. As we fine-tune later and later models, the distance traveled away from the trajectory is smaller and smaller, i.e., changes in the representation are smaller. For a fully-trained CIFAR-10 model which interpolates the training data, the distance traveled by fine-tuning is very small (the points are almost indistinguishable in the picture); this is because both $P^1$ and $P^2$ are subsets of CIFAR-10.

Algorithms for transfer learning train on a source task before fine-tuning the model on the target task. If two tasks share a large part of their training trajectory, then we may start the fine-tuning from many shared intermediate points—there are many such points. If the chosen point is farther along

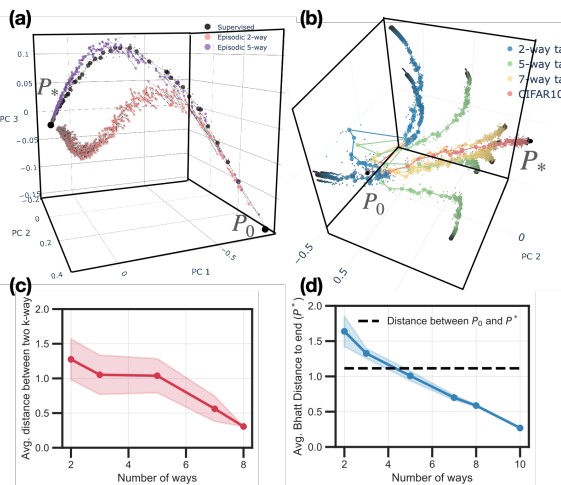

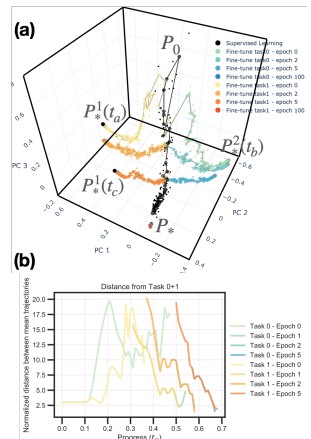

**Figure 2: (a)** Training trajectories for supervised learning (black), 2-way (pink) and 5-way episodic meta-learning (purple). Trajectories of 5-way meta-learning are very similar to those of supervised learning and eventually reach very similar models and high test accuracy. In contrast, 2-way meta-learning has a much longer trajectory (about $40\times$ longer in Riemann length than black) and does not reach a good test accuracy (on all 10 CIFAR-10 classes). Representations are similar during early parts of training even if these are quite different learning mechanisms. **(b)** Trajectories of 2-way (blue), 5-way (green), 7-way (yellow) meta-learning compared to supervised learning (red). For large "way", trajectories are similar to supervised learning but they deviate from the red trajectories quickly for small ways. **(c)** Average distance between two $k$-way meta-learning trajectories decreases with $k$, this is a geometric evidence of the variance of predictions of learned representations. **(d)** Training with a small way leads to models that predict poorly on test data (large distances from truth). These embeddings were calculated using the training dataset. The rationale being that we wanted to show how different meta-learning and supervised learning are during training.

**Figure 3: (a)** Fine-tuning trajectories on Airplane vs. Automobile, and Bird vs. Cat sub-tasks of CIFAR-10 (warm and cold hues) pre-trained from different points along the trajectory of supervised learning. If the pretrained model has progressed further towards the truth $P_*$, then fine-tuning it on a sub-task does not change the representation much. The final trajectory (fine-tuning from epoch 100) is indistinguishable from $P_*$. **(b)** Bhattacharyya distance between the mean trajectories normalized by the average of the tube radii (like Fig. 1g). Models (say, fine-tuned after Epoch 5 on Task 1) go *backwards* in progress, i.e., they unlearn the pretrained representation in order to fit the new task. This occurs as early as Epoch 1 here. It suggests that learning occurs extremely rapidly at the beginning and determines the efficiency of fine-tuning. Some curves here are not visible because they are overlapping heavily.

in terms of progress then the efficiency resulting from using the source task is higher because the trajectory required to fit the target task is shorter; such trajectories were used in (Gao & Chaudhari, 2021) to define a distance between tasks. As we saw in Result 2, trajectories of different tasks bifurcate after a shared part. The resultant deviation less for related tasks and more for dissimilar tasks (Fig. 3a, Fig. 1a,f). Therefore it is difficult to know *a priori* from which point one should start the fine-tuning from without knowing the manifold of the target task. In particular, our geometric picture indicates that fine-tuning from a fully-trained model can be detrimental to the accuracy on the target task. This has been noticed in a number of places in the transfer learning literature, e.g., Li et al. (2020), and has also been studied theoretically (Gao & Chaudhari, 2020).

**Result 5. Episodic meta-learning algorithms fit a similar model eventually as that of supervised learning, even if the two traverse very different trajectories during training.** Meta-learning methods build a representation which can be adapted to a new task (Thrun & Pratt, 2012). We studied a common variant, the so-called episodic training methods (Bengio et al., 1992), through the perspective of few-shot learning (Vinyals et al., 2016). In these methods, each mini-batch consists of samples from $C^w$ classes (called "way", chosen randomly out of the $C$ classes) split into two parts: a "support set" $D_s$ of $s$ samples/class (called "shot"), and a "query set" $D_q$ of $q$ samples/class. Typical methods, say prototypical networks of Snell et al. (2017b), implement a clustering loss on features of the query samples using the averaged features of the support samples $\varphi_c = s^{-1} \sum_{\{x \in D_s, y^*(x)=c\}} \varphi(x; w_1)$ for all $c = 1, \ldots, C^w$ as the cluster centroids. This involves maximizing the likelihood

$$\forall x \in D_q, \text{ if } y^*(x) = c : p(\varphi(x; w_1)) \propto \exp\left(-\|\varphi_c - \varphi(x; w_1)\|^2/2\right) \qquad (11)$$

over query samples. If features $\varphi$ lie on an $\ell_2$ ball of radius 1, then maximizing this likelihood is akin to maximizing the cosine similarity between cluster centroids computed from the support samples and features of the query samples. At the end of training, the same clustering loss with the learned backbone $w_1$ is used to predict on unseen classes (using "few" support samples to compute centroids).

We compared trajectories of episodic meta-learning to the trajectory taken by supervised learning using the cross-entropy loss over all $C$ classes in Fig. 2. Episodic meta-learning methods are quite different from supervised learning so it seems surprising that both arrive at the same solution; see Fig. 2a,b and Fig. A3 for distances between trajectories. But this is consistent with recent literature which has noticed that the performance of few-shot learning methods using supervised learning (followed by fine-tuning) is comparable to or better than episodic meta-learning (Dhillon et al., 2020; Kolesnikov et al., 2020; Fakoor et al., 2020). The supervised learned representation also minimizes the clustering loss (11). We find that the trajectory length of episodic training is about $40\times$ longer than that of supervised learning.

Few-shot accuracy of episodic training is better with a large way (Gidaris & Komodakis, 2018). We trained models with different ways to study this (Fig. 2b). We find that the radius of the tube that encapsulates the models of 2-way meta-learning around their mean trajectory is very large, almost as large as the total length of the trajectory, i.e., different models trained with a small way learn very different representations. Tube radius decreases as the way increases (Fig. 2c). Further, the distance of models from the truth $P_*$ (which is close to the end point of the supervised learning model) is higher for a small way (Fig. 2d). This is geometric evidence of the widely used empirical practice of using a large number of way in episodic meta-learning. Observe that as the way increases, the trajectory of the episodic meta-learning becomes more and more similar to that of supervised learning.

**Result 6. Contrastive learning methods trained on different datasets learn similar representations**
Contrastive learning (Becker & Hinton, 1992) learns representations without using ground-truth labels (Gutmann & Hyvärinen, 2010; Chen et al., 2020a). It has been extremely effective for self-supervised learning (Doersch & Zisserman, 2017; Kolesnikov et al., 2019), e.g., prediction accuracy with 1–10% labeled data is close to that of supervised learning using all data (Chen et al., 2020b). We compared representations learned using contrastive learning with those from supervised learning to understand some aspects of why the former are so effective.

Consider a task $P$ and a set of augmentations $G$ (e.g., cropping, resizing, blurring, color/contrast/brightness distortion etc.). Given inputs (say images) $x$ from $P$, contrastive learning forces the representation $\varphi(g(x); w_1)$ and $\varphi(g'(x); w_1)$ (shortened to $\varphi(g(x))$ below) of the same input for two different augmentations $g, g'$ to be similar. And forces it to be different from representations of other augmented inputs $x'$ (Zbontar et al., 2021; Bachman et al., 2019; Dosovitskiy et al., 2014).

We used a popular method called SimCLR (Chen et al., 2020a) to perform contrastive learning on images from four sets of classes (airplane-automobile, bird-cat, ship-truck) and all of CIFAR-10. We compared the learned representation to that from supervised learning on two tasks (airplane-automobile and all of CIFAR-10) in Fig. 4. Models trained using contrastive learning on two-class datasets learn very different representations from models trained on the same task but using supervised learning. Models trained using contrastive learning on different datasets learning similar representations (trajectories of all three two-class datasets are very close to each other). This is reasonable because contrastive learning does not use any information from the labels. It is surprising however that the trajectory of models from contrastive learning on these two-class datasets is similar to trajectories of models from contrastive learning on the entire CIFAR-10.

We also performed an experiment where we compare the representations of semi-supervised (Fixmatch (Sohn et al., 2020)), contrastive (SimCLR (Chen et al., 2020a), Barlow-twins (Zbontar et al., 2021)) and supervised learning; see Appendix G. All three trajectories are similar to the trajectory of supervised learning. We find that the trajectory for semi-supervised learning deviates from the supervised learning trajectory initially, but the two are very similar for larger values of progress ($t_w$).

## 4    RELATED WORK AND DISCUSSION

**Understanding the space of learning tasks.** A large body of work has sought to characterize relationships between learning tasks, e.g., domain specific methods (Zamir et al., 2018; Cui et al., 2018; Pennington et al., 2014), learning theoretic work (Baxter, 2000; Maurer, 2006; Ben-David et al., 2010; Ramesh & Chaudhari, 2022; Tripuraneni et al., 2020; Hanneke & Kpotufe, 2020; Caruana, 1997), random matrix models (Wei et al., 2022), neural tangent kernel models (Malladi et al., 2022) and information-theoretic analyses (Jaakkola & Haussler, 1999; Achille et al., 2019a;b). Broadly speaking, this work has focused on understanding the accuracy of a model on a new task when it is

trained upon a related task, e.g., relationships between tasks are characterized using the excess risk of a hypothesis.

Like these above works, our methods also allow us to say things like "task $P^1$ is far from $P^2$ as compared to $P^3$". But we heavily exploit information geometry and can go much further. Our methods shed light on the geometric structure in the space of tasks using the geometry of probabilistic models of these tasks. This allows us to make quantitative conclusions such as "the divergence between $P^1$ and $P^2$ eventually is more than that of $P^1$ and $P^3$, but representations learned on these tasks are similar for 30% of the way". Therefore our methods allow us to paint a picture of the space of the tasks that is quantitative and globally consistent.

There is strong structure in typical inputs, e.g., recent work on understanding generalization (Yang et al., 2022; Bartlett et al., 2020) as well as older work such as (Simoncelli & Olshausen, 2001; Field, 1994; Marr, 2010) has argued that visual data is effectively low-dimensional. Our works suggests that tasks also share a low-dimensional structure. Just like the effective low-dimensionality of inputs enables generalization on one task, effective low-dimensionality of the manifold of models trained on different tasks could perhaps explain generalization to new tasks.

**Relationships between tasks in computational neuroscience.** Our results are conceptually close to those on organization and representation of semantic knowledge (Mandler & Mc-Donough, 1993). Such work has primarily used simple theoretical models, e.g., linear dynamics of Saxe et al. (2019) (who also use MDS). Our tools are very general and paint a similar picture of ontologies of complex tasks. There is some resemblance in how our different networks learn the task in a similar fashion (low-dimensional manifold, trajectories across different random seeds and similar tasks are very close to each other), and how different individuals share the same notion of what constitutes a task (Rosch & Mervis, 1975). Concept formalization and

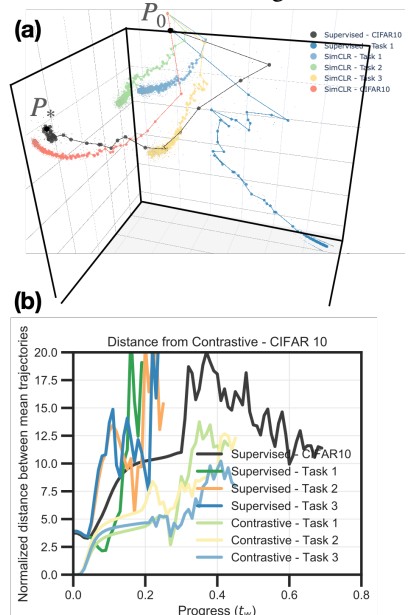

**Figure 4: (a)** Trajectories of contrastive learning (SimCLR) on 3 datasets (two classes each) and entire CIFAR-10 compared to those of supervised learning. SimCLR on entire CIFAR-10 learns a similar representation as that of the supervised learned model $P_*$ (which fits the training data perfectly). SimCLR trajectories are close to each other even if different datasets were used to train them. It may seem from the embedding that SimCLR trajectories are similar to that supervised learning, which would be very surprising because the former does not use any labels, but see below. **(b)** Bhattacharyya distance between the mean trajectories of all models and the mean trajectory of SimCLR on all CIFAR-10. This distance is normalized by the average of the tube radii (like Fig. 1g). SimCLR trajectories of two-class datasets are indeed very close to each other (mean distance is $\sim 5\times$ more than their tube radii for about 45% of the way ($t_w \approx 0.2$)). This plot indicates that two-class SimCLR trajectory (light blue) is close to SimCLR on all of CIFAR-10. But two-class supervised learning trajectory (darker blue) is much farther away from SimCLR on all of CIFAR-10.

specialization over age (Vosniadou & Brewer, 1992) also resembles our experiment in how fine-tuning models trained for longer periods changes the representation marginally. Our goals are similar to those of works like Sorscher et al. (2021) but our focus is very different. We have focused our techniques on studying how representation learning on multiple tasks works.

**Visualizing training trajectories of deep networks.** A large number of works have investigated trajectories of deep networks and the energy landscape during or after training using dimensionality reduction techniques (Horoi et al., 2021; Li et al., 2018; Huang et al., 2020) and study if learning occurs in a low-dimensional sub-space (Gur-Ari et al., 2018; Antognini & Sohl-Dickstein, 2018). The key distinction with respect to this existing work—and our key technical innovation—is that we visualize the function space, not the weight space. We study the function space of deep networks by evaluating the underlying probabilistic model on finite samples. While the weight space has symmetries (Freeman & Bruna, 2017; Garipov et al., 2018) and nontrivial dynamics (Tanaka & Kunin, 2021; Chaudhari & Soatto, 2018), the function space, i.e,. our large vector $[0, 1]^{N \times C} \ni \{p_w(c \mid x_i)\}$ completely characterizes the output of the model. In comparison, the loss or the error which are typically used to reason about relationships between tasks, are very coarse summaries of the predictions. Any two

models trained on the same task can be rigorously studied in our "finite-dimensional function space" to understand their predictions. Another key innovation is that our mapping procedure allows us to study the representation of the penultimate layer of any two models, even if they were trained on different tasks.

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

## A    DETAILS OF THE EXPERIMENTAL SETUP

**Data**
We performed experiments using two datasets.

1. CIFAR10 (Krizhevsky, 2009) has 10 classes (airplane, automobile, bird, cat, deer, dog, frog, horse, ship, truck) with RGB images of size 32×32, and

2. Imagenet (Deng et al., 2009) has 1000 classes each with about 1000 RGB images of size 224×224.

**Architectures**    We use a Wide-Resnet (Zagoruyko & Komodakis, 2016) architecture for supervised learning experiments on CIFAR-10 (WRN-16-4 with depth 16 and widening factor of 4) and a Resnet-18 (He et al., 2016) to train a model using SimCLR. All experiments on Imagenet use the Resnet-50 architecture.

All convolutional layers are initialized using the Kaiming-Normal initialization. For the Wide-Resnet, the final pooling layer is replaced with an adaptive pooling layer in order to handle input images of different sizes.

We make three modifications to these architectures.

1. We remove the bias from the final classification layer; this helps keep the logits of the different tasks on a similar scale.

2. In the experiments pertaining to Results 4 and 5, we replace batch normalization with layer norm in the Wide-Resnet. This is because we found in preliminary experiments that batch-normalization parameters make training meta-learning models very sensitive to choices of hyper-parameters (e.g., the support or query shot), and that the learned representations of new tasks were quite different in terms of their predictions (and thereby the Bhattacharyya distance) but all the difference was coming from modifications to the BN parameters.

3. In the Resnet-50, we replace the pooling layers with BlurPool (Zhang, 2019). The bias parameter in batch normalization is set to zero with the affine scaling term set to one.

**Training procedure**    All models are trained in mixed-precision (32-bit weights, 16-bit gradients) using stochastic gradient descent (SGD) with Nesterov's acceleration with momentum coefficient set to 0.9 and cosine annealing of the learning rate schedule. Batch-normalization parameters are excluded from weight decay.

CIFAR10 datasets use padding (4 pixels) with random cropping to an image of size 28×28 or 32×32 respectively for data augmentation. CIFAR10 images additionally have random left/right flips for data augmentation. Images are finally normalized to have mean 0.5 and standard deviation 0.25.

Supervised learning models (including fine-tuning) for CIFAR10 are trained for 100 epochs with a batch-size of 64 and weight decay of $10^{-5}$ using the Wide-Resnet.

Episodic meta-learners are trained using a Wide-Resnet and with the prototypical loss (Snell et al., 2017a). For the 2-way meta-learner, each episode contains 20 query samples and 10 support samples. For the 5-way meta-learner, each episode contains 50 query samples and 10 support samples. We found Result 5 to hold across different choices of these hyper-parameters in small-scale experiments. Models are trained for around 750 epoch and the episodic learner is about 5 times slower to train with respect to wall-clock time.

We train models using SimCLR on CIFAR10 and on tasks created from CIFAR10. For the augmentations, we use random horizontal flips, random grayscale, random resized crop and color jitter. Models are trained for 200 epochs for 2-way classification problems and for 500 epochs when trained on the entirety of CIFAR10 with the Adam optimizer and an initial learning rate of 0.001.

### A.1    EXPERIMENTS ON IMAGENET

We make use of FFCV (Leclerc et al., 2022). which is a data-loading library that replaces the pytorch Dataloader. FFCV reduces the training time on Imagenet to a few hours, which allows us to train 100s of models on Imagenet, or on tasks created from it. Our implementation of Imagenet training builds on the FFCV repository [1].

Imagenet models are trained for 40 epochs with progressive resizing – the image size is increased from 160 to 224 between the epochs 29 and 34. Models are trained on 4 GPUs with a batch-size of 512. The training uses two types of augmentations – random-resized crop and random horizontal flips. Additionally, we use label smoothing with the smoothing parameter set to 0.1.

---

[1] https://github.com/libffcv/ffcv-imagenet/tree/main/

## A.2 IMPLEMENTING INPCA IN VERY HIGH DIMENSIONS

We calculate an InPCA embedding of models along multiple trajectories, e.g., a typical experiment has about 25 trajectories (multiple random seeds, tasks, or representation learning methods) and about 50 models (checkpoints) along each trajectory. Each model is a very high-dimensional object (with dimensionality $NC$ where $N \sim 10^5$ and $C \sim 10\text{-}10^3$). Even if the matrix $D$ in (9) is relatively manageable with $n \sim 1250$, each entry of $D$ is $\mathrm{d_B}(P_u, P_v)$ and therefore requires $\sim 10^8$ operations to compute. Implementing InPCA—or even PCA—for such large matrices requires a large amount of RAM. We reduced the severity of this issue to an extent using Numpy's memmap functionality https://numpy.org/doc/stable/reference/generated/numpy.memmap.html. Also note that calculating only the top few eigenvectors of (9) suffices to visualize the models, we do not need to calculate all.

The formula (2) is an effective summary of the discrepancies between how the predictions made by two probabilistic models differ; even small differences in two models, e.g., even if both $P_u$ and $P_v$ make mistakes on exactly the same input samples, if $p_u^n(c)$ is slightly different than $p_v^n(c)$ for even one of $n$ or $c$, the divergence is non-zero. InPCA is capable of capturing the differences between two such models (9). However, when the number of classes is extremely large, the number of terms in the summation is prohibitively large and analyzing the discrepancies or calculating the embedding becomes rather difficult.

We also developed a method to work around this issue. We can use a random stochastic matrix (whose columns sum up to 1) to project the outputs for each sample $\{p_u^n(c)\}_{c=1,\ldots,C}$ into a smaller space before calculating (2). This amounts to pretending as if the model predicts not the actual classes but a random linear combination of the classes (even if the model is trained on the actual classes). This is a practical trick that is necessary only when we are embedding a very large number of very high-dimensional probabilistic models. We checked in our Imagenet experiments that using this trick gives the same embeddings.

In this paper, we did not need to use this projection trick.

## B CALCULATING MEAN TRAJECTORIES

We defined the distance between two trajectories to be $\mathrm{d_{traj}}(\tau_u^{1\to\mathrm{U}}, \tau_v^{2\to\mathrm{U}})$, i.e., the integral of the Bhattacharyya distance between the trajectories after mapping them to the same task and re-indexing them using the geodesic. This quantity is key in our work. Say we wish to compare a model trained on two tasks: cats vs. dogs and airplane vs. truck from CIFAR-10. We initialize multiple models for each of these two supervised learning problems (and we do so for every experiment in this paper) and train these 10 models. We can now calculate the mean trajectory of models on a task

$$\operatorname*{argmin}_{\tau_\mu^1} \frac{1}{K} \sum_{k=1}^{K} \mathrm{d_{traj}}(\tau_{u_k}^1, \tau_\mu^1).$$

This optimization problem is very challenging because the variable is a trajectory of probabilistic models in a high-dimensional space. Even if we were to split this minimization and do it independently across time, this is still difficult because the solution is the so-called Bhattacharyya centroid on the product manifold defined in (1) and cannot be computed in closed form. See (Nielsen & Boltz, 2011) for an iterative formula. We therefore simply take the arithmetic mean of the probability distributions, i.e., $P_{\mu(t)} = \frac{1}{K} \sum_{k=1}^{K} P_{w_i(t)}$. This is similar to ensembling. We use the radius of the tube around the mean trajectory, i.e.,

$$r_u = \max_k \mathrm{d_{traj}}(\tau_{u_k}^1, \tau_\mu^1)$$

to normalize distances (more precisely, we normalize using the average of the radii of the two trajectories being compared). Note that this radius depends upon time (and is computed after mapping and reindexing the trajectories). If the distance between the means of two sets of trajectories is smaller than their individual average radii, then the tubes around the means intersect each other. In such cases, one can say that the representations learned (at that time point) are not distinguishable. We next show all distances between reindexed points along the trajectories discussed in Figs. 1, 3 and 4. Note that each curve gives the integrands in (5), not the integral.

## C BHATTACHARYYA DISTANCE

In this section, we provide additional details regarding (2). Let $\bar{y} = (y_1, \cdots y_N)$, denote the labels assigned to each of the $N$ samples. Since there are $C$ classes in total, $\bar{y}$ can take a total of $N^c$ different values denoted by the set $Y^C$. Given, two models $P_u$ and $P_v$, the Bhattacharyya averaged over the samples is

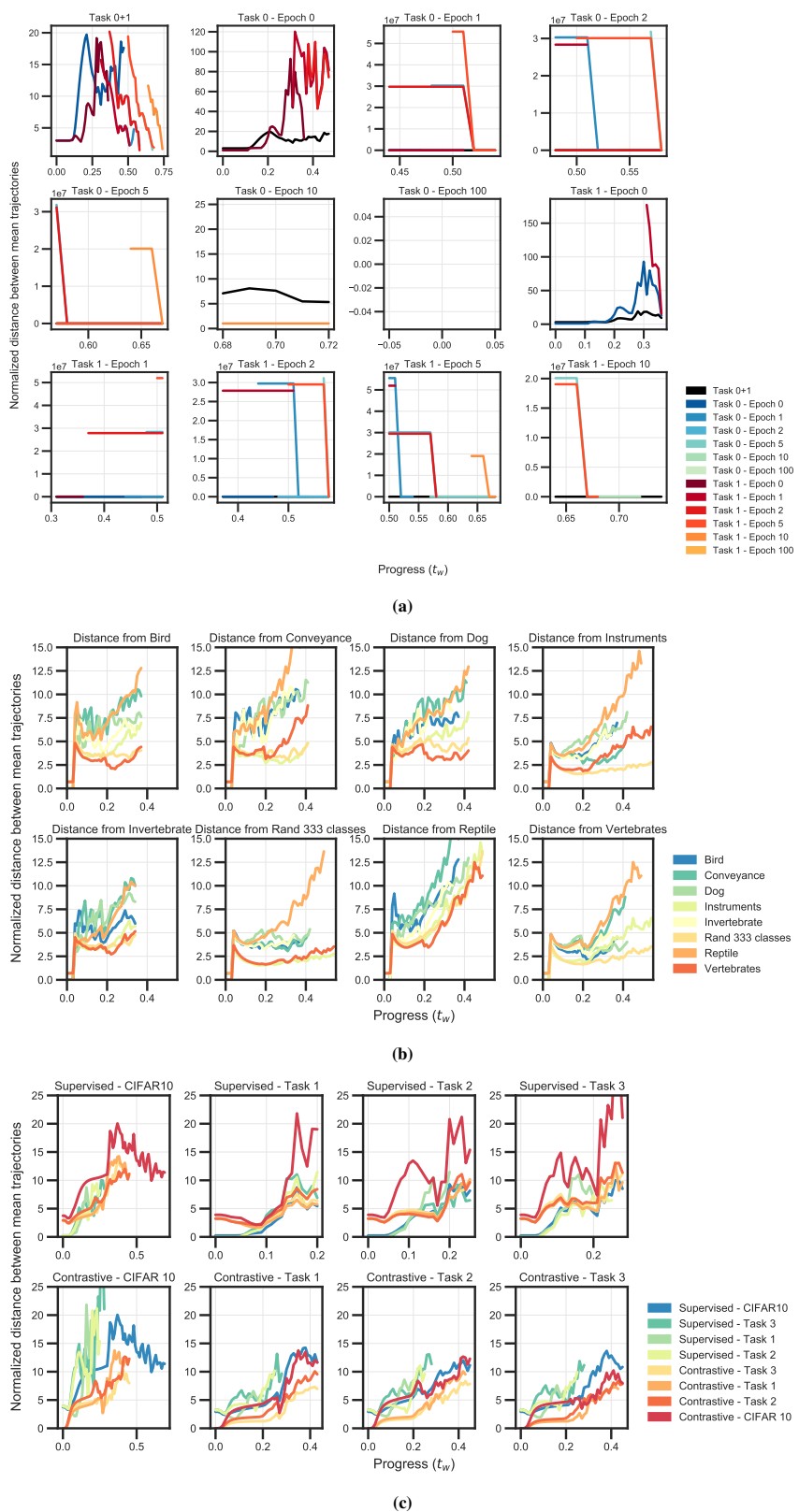

**Figure A1:** This figure shows the extended version of the distances between trajectories of probabilistic models; two of them are identical to the ones in Fig. 3b and Fig. 4b.

$$
\begin{aligned}
d_{\mathrm{B}}(P_u, P_v) &:= -N^{-1} \log \left( \sum_{\bar{y} \in Y^C} \sqrt{P_u(\bar{y}) P_v(\bar{y})} \right) \\
&= -N^{-1} \log \left( \sum_{\bar{y}} \prod_{n=1}^{N} \sqrt{p_u(y_n)\, p_v(y_n)} \right) \\
&= -N^{-1} \log \left( \sum_{y_1=1}^{C} \sum_{y_2=1}^{C} \cdots \sum_{y_N=1}^{C} \left( \prod_{n=1}^{N} \sqrt{p_u(y_n)\, p_v(y_n)} \right) \right) \\
&= -N^{-1} \log \left( \sum_{y_1=1}^{C} \cdots \sum_{y_{N-1}=1}^{C} \left( \prod_{n=1}^{N-1} \sqrt{p_u(y_n)\, p_v(y_n)} \right) \left( \sum_{y_N=1}^{C} \sqrt{p_u(y_N)\, p_v(y_N)} \right) \right) \\
&= \qquad \vdots \\
&= -N^{-1} \log \left( \sum_{y_1=1}^{C} \cdots \sum_{y_{N-k}=1}^{C} \left( \prod_{n=1}^{N-k} \sqrt{p_u(y_n)\, p_v(y_n)} \right) \prod_{\substack{i= \\ N-k+1}}^{N} \left( \sum_{y_i=1}^{C} \sqrt{p_u(y_i)\, p_v(y_i)} \right) \right) \\
&= \qquad \vdots \\
&= -N^{-1} \log \left( \prod_{i=1}^{N} \left( \sum_{y_i=1}^{C} \sqrt{p_u(y_i)\, p_v(y_i)} \right) \right) \\
&= -N^{-1} \sum_{i=1}^{N} \log \left( \sum_{y_i=1}^{C} \sqrt{p_u(y_i)\, p_v(y_i)} \right)
\end{aligned}
$$

## D  MEASURING GOODNESS-OF-FIT OF AN INPCA EMBEDDING USING EXPLAINED STRESS

We would like to measure if a $k$-dimensional sub-space accurately preserves the true distances. For this purpose, we define a quantity called the "explained stress" that estimates the fraction of pairwise distances in the original space that are preserved in the $k$-dimensional embedding. This is analogous to the explained variance in principal component analysis (PCA); but explained variance is a measure of the how well the original points are preserved in the embedding whereas explained stress approximates how well pairwise Bhattacharyya distances are preserved. If we consider the embedding to be given by first $k$ eigen-vectors, then the explained stress ($\chi_k$) is

$$
\chi_k = 1 - \frac{\left\| W - \sum_{i=1}^{k} \Sigma_{ii}\, U_i U_i^{\top} \right\|_{\mathrm{F}}}{\|W\|_{\mathrm{F}}} = 1 - \sqrt{\frac{\sum_{i=k+1}^{m} \Sigma_{ii}^2}{\sum_{i=1}^{m} \Sigma_{ii}^2}}. \tag{12}
$$

Note that InPCA finds an embedding that exactly maximizes the objective $\chi_k$.

## E  IMPRINTING RESULTS IN WEIGHTS THAT MAXIMIZE THE LOG-PROBABILITY UNDER A NORM CONSTRAINT

Consider a total of $C$ classes. In this section, we show that imprinting results in weights $\{w_c\}_{c=1}^{C}$ that maximize the log-probability of the samples while satisfying the constraint $||w_c|| = 1$. Hence, imprinting can be used instead of training the last layer.

Let the set of all samples in class $c$ be $\{x_i^c\}_{i=1}^{n_c}$. We define the log-probability with which sample $x_i$ belongs to $c$, as $\omega_c \cdot \varphi(x_i)$, where $\varphi(x_i)$ are the feature of sample $x_i$. Hence,

$$
\sum_{i=1}^{n_c} \log p(y = c \mid x_i^c) = \sum_{i=1}^{n_c} w \cdot \phi(x_i^c) = w_c \cdot \sum_{i=1}^{n_c} \varphi(x_i^c). \tag{13}
$$

If we don't have a constraint on the norm of $w_c$, then we can maximize (13) by sending $w_c \to \infty$. Let

$$
w_c^* = \operatorname*{argmax}_{w: ||w||=1} w_c \cdot \sum_{i=1}^{n_c} \varphi(x_i^c),
$$

denote the weights that maximize the log-probability under the $\ell_2$ norm constraint. Since $w = x/\|x\|$ maximizes the objective $w \cdot x$, subject to the constraint that $\|w\| = 1$, the optimal value for $w_c$ is $\sum_i \phi(x_i^c)/\|\sum_i \phi(x_i^c)\|$.

## F   COMPARING DIFFERENT SEMI-SUPERVISED, CONTRASTIVE AND SUPERVISED LEARNING METHODS

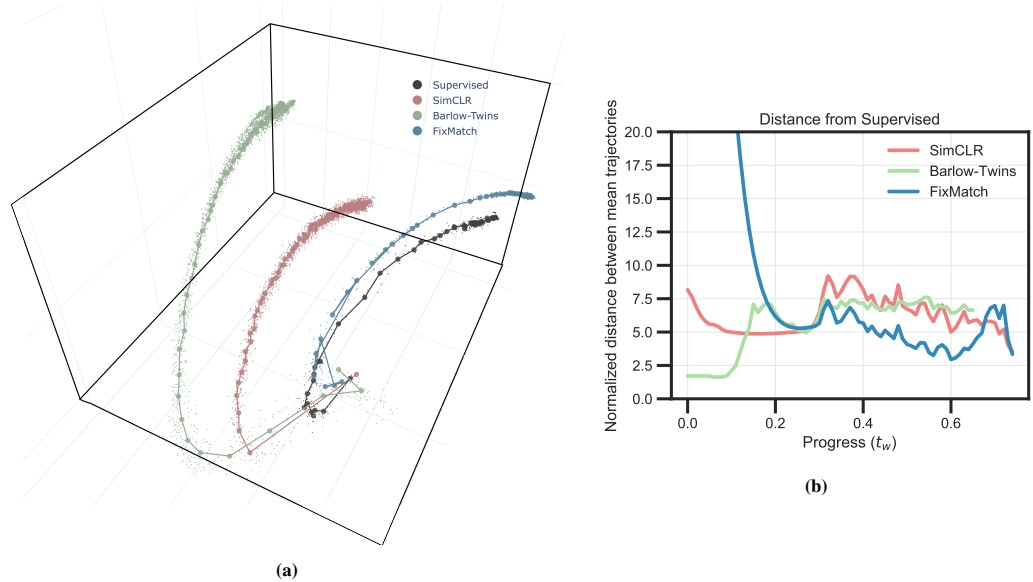

(a)

(b)

**Figure A2:**   We consider 4 methods for training on CIFAR10: supervised learning, SimCLR (Chen et al., 2020a), Barlow-twins (Zbontar et al., 2021) and Fixmatch (Sohn et al., 2020). Fixmatch is a semi-supervised learning method and has access to 2500 labeled samples in addition to 47500 unlabeled samples. SimCLR and Barlow-twins are contrastive learning methods that use all the 50,000 unlabeled samples for training.
**(a)** We plot the trajectories for supervised, semi-supervised and contrastive learning. **The trajectory of semi-supervised learning (Fixmatch) eventually resembles supervised learning in comparison to contrastive learning methods.** All methods result in remarkably similar trajectories although some of these methods are clearly trained using only unlabeled data.
**(b)** We plot the normalized distance of trajectories corresponding to contrastive and semi-supervised learning to the trajectory of supervised learning. **Semi-supervised learning (Fixmatch) deviates considerably from the other methods at the beginning**. We speculate that this is because the trajectory of Fixmatch is influenced by the 2500 labeled samples. As, training progresses, Fixmatch becomes increasingly similar to supervised learning as evidenced by the dip in the blue line for larger values of progress $(t_w)$.

## G   DISTANCE BETWEEN TRAJECTORIES OF EPISODIC META-LEARNING AND SUPERVISED LEARNING

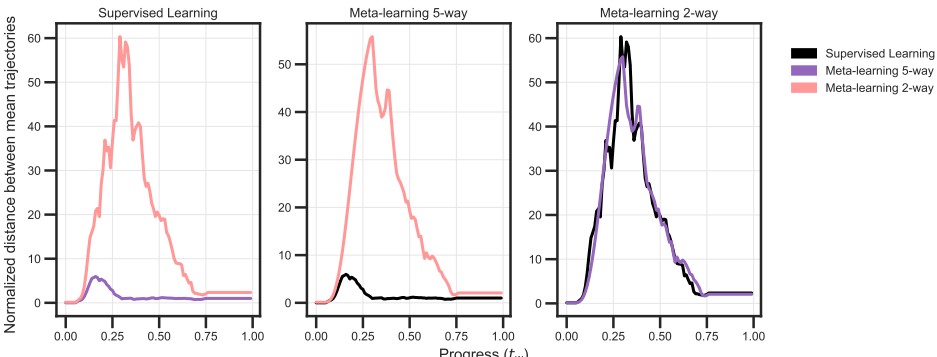

**Figure A3:** **Distance between trajectories of supervised and meta-learning at different values of progress.** Distances between the average trajectories of different algorithms (e.g., 2-way episodic learning and supervised learning, and 5-way episodic learning and supervised learning in the leftmost panel) are normalized by the average of the radii of the tubes corresponding to each trajectory. We find that trajectories of 2-way meta-learning deviate significantly from those of supervised learning for a large fraction of the trajectory. On the other hand, 5-way meta-learning is similar to the supervised learning trajectory for almost the entirety of the trajectory.

