# OpenReview forum: "A Picture of the Space of Typical Learning Tasks"
_ICLR.cc/2023/Conference — Submitted to ICLR 2023_

### Official Review · Reviewer_hj5S · 2022-10-24

**Confidence:** 3
**Correctness:** 4
**Technical Novelty And Significance:** 3
**Empirical Novelty And Significance:** 3
**Recommendation:** 6

**Clarity, Quality, Novelty And Reproducibility:**

- The paper is generally clear and easy to follow. Some notions in the paper such as "Progress" are not defined before being used.
- The main novelty of the paper seems to be quantitative metric to computer distance between mean trajectories.


**Strength And Weaknesses:**

Strengths:
- I generally find this sort of analysis interesting which can help us better understand what goes on the learning process.
- Having a quantitative method to compare learning trajectories is very interesting.
- The paper is generally organized well. The experiments support the contribution claims.

Weakness:
- The authors mention in the abstract that they have surprising findings, however, all their findings are inline with the literature.
- They could apply their method on more diverse dataset instead of sampling classes from the same dataset or 2 relatively similar datasets (i.e. CIFAR10 and Imagenet)
- CIFAR10 is a very small dataset to apply self-supervised learning.
- It would have been more interesting if they had conducted experiments on more self-supervsing methods and to compare different self-superviseing methods.
- Fig 4 does not seem to be referenced in the text.
- Organization of results is very condense. It could be done better.


**Summary Of The Paper:**

This paper analyses representations learned by neural network. The analysis is done via isometric mapping of the probabilistic model into a lower dimensional maniforld such that it preserves the pairwise distance between models. They propose a distance metric to compare different learning trajectories.  They also use InPCA to embed the model to a lower dimensional space. By doing so they can visualize a trajectory of the learning process and compare trajectories of different models.
The main findings are: 1)the manifold of  the models trained on different tasks are low-dimensional 2)supervised learning help training of dissimilar tasks especially if it has diverse classes. 3) pre-training for longer epochs could hinder fine-tuning of the downstream task. 4) episodic meta-learning fits similar models than supervised learning even if they take different trajectories. 5) contrastive learning trained on different datasets learn similar representations.


**Summary Of The Review:**

I think the paper has merits for publication. Analysis of deep learning representations and models can help us better understand what goes on the learning process. It seems to me the main novelty of the paper is distance metric between training trajectories which can help us measure similarity of models and tasks. The paper could have performed more extensive experiments (see weaknesses for details. )

---

> ### Author Response · Authors · 2022-11-17
> **Response to Reviewer hj5s (1/2)**
>
> Thank you for the comments on the manuscript. We appreciate your comments and have hopefully addressed some of your concerns. We have added additional experiments on semi-supervised and self-supervised learning.
>
> **>>> The authors mention in the abstract that they have surprising findings, however, all their findings are inline with the literature.**
>
> We have been diligent in citing related work and in making quantitative and precise claims using sophisticated mathematical techniques. In the following rebuttal, we have provided a detailed reply to your comments. It is of course possible that we may have missed something. We will be very happy to investigate and cite work that has made these claims precisely if the reviewer points us to it.
>
> **>>> They could apply their method on more diverse dataset instead of sampling classes from the same dataset or 2 relatively similar datasets (i.e. CIFAR10 and Imagenet)**
>
> For Results 1-3, we have used Imagenet to create diverse tasks (all of Imagenet, 333 random classes, Instruments, Vertebrates, Dogs, Reptiles, Birds, Conveyance, Invertebrates). This presents a huge diversity in terms of the number of classes, number of images, and the relatedness of the classes. For Results 4-6, we have used the CIFAR-10 dataset and many different representation learning techniques (fine-tuning on different tasks from different checkpoints, few-shot learning with different ways, multiple contrastive learning methods, and now semi-supervised learning methods). Some of our experiments are extremely large (e.g., for embedding Imagenet, we have written code to implement these complex ideas from information geometry at a scale that has never been done before (see Appendix A.2)).
>
> Altogether, this paper already presents extensive evidence for six different results (seven including the new experiments on semi-supervised learning) that involve very different experimental settings. This paper proposes very general set techniques and demonstrates their effectiveness by revealing some remarkable phenomena. The space of tasks is enormous and these techniques can certainly be applied to many other tasks to discover many other interesting properties of tasks and algorithms. But this is best left for future work.
>
> **>>> CIFAR10 is a very small dataset to apply self-supervised learning.**
>
> Self-supervised learning models have also been trained on CIFAR-10 by a number of papers [4,5,6]. Contrastive learning has also been applied to smaller datasets in the context of unsupervised domain-adaptation [7]. In our experiments, SimCLR trains to an accuracy of 87.36% on CIFAR10 (after imprinting the final model), which is only ~7% worse than a typical run of supervised learning in spite of not using any labels whatsoever.
>
> **We chose CIFAR-10 for contrastive learning because it is incredibly expensive to train on Imagenet.** Zbontar et al. [1] report that their method requires **32 V100 GPUs over 124 hours**. For all experiments in our paper, we train on 5 random seeds (to create trajectories from different initializations, interpolate them using geodesics, compute averages etc.). An experiment on Imagenet would be enormously expensive **(about $60,000 on AWS for 5 seeds)**. **Please note that this would be the cost of a sub-part of Result 6; there are 5 other results in the paper.**
>
> **>>> It would have been more interesting if they had conducted experiments on more self-supervsed methods and to compare different self-supervised methods.**
>
> Thank you for the suggestion. We have added an experiment that compares semi-supervised (Fixmatch[2]), self-supervised learning (SimCLR[3], Barlow-twins[1]) and supervised learning to Appendix F of the manuscript. We refer to this experiment in Result 6.
>
> Interestingly, we find that all 3 trajectories are similar to the trajectory of supervised learning. **While we know that the final accuracy of each of the methods are similar, it is surprising to find that the trajectories of the methods are also similar.** Additionally, we find that semi-supervised learning deviates from the supervised learning trajectory initially, but the two are very similar for larger values of progress (t_w).

---

> ### Author Response · Authors · 2022-11-17
> **Response To Reviewer hj5s (2/2)**
>
> **>>> The paper could have performed more extensive experiments (see weaknesses for details. )**
>
> We have included a new experiment on semi-supervised learning and one new self-supervised learning method; please see the response to the previous comment.
>
> We implore the reviewer to consider the scale of the experiments in this manuscript already. We are not making claims on toy datasets or on the basis of weak experiments. We have used nine different tasks from Imagenet with very dissimilar classes, many sub-tasks on CIFAR-10 and five different representation learning methods (supervised, fine-tuning, episodic, contrastive and semi-supervised learning). For each of these, we have trained on multiple random seeds and conducted rigorous analysis of the representations learned by these methods to glean precise quantitative conclusions. All this, in addition to developing entirely new theoretical and computational information geometry techniques.
>
> **These experiments are already quite extensive and took about 30,000 GPU-hours (without counting time spent to develop the theory). This comes to about $75,000 on AWS. To emphasize, Figure 1a alone requires about 50+ training runs on Imagenet.**
>
> **These remarkable phenomena touch so many established fields. So** **we do appreciate your curiosity.** **If there is a specific experiment that you propose and there is existing evidence to suggest that our claims will not hold for it, we will happily modify the claim and/or conduct the experiment to check.**
>
>
> **>>> Some notions in the paper such as "Progress" are not defined before being used.**
>
> Equation (4) defines progress. We have made it bold now. We had used the word informally in the introduction (bullet 2) which we have clarified now.
>
> **>>> Fig 4 does not seem to be referenced in the text.**
>
> Thank you, this is fixed now.
>
>
> **References**
>
> 1. Zbontar, Jure, et al. "Barlow twins: Self-supervised learning via redundancy reduction." ICML 2020
> 2. Sohn, Kihyuk, et al. "Fixmatch: Simplifying semi-supervised learning with consistency and confidence." NeurIPS 2020.
> 3. Chen, Ting, et al. "A simple framework for contrastive learning of visual representations." ICML 2020.
> 4. Shen, Zhiqiang, et al. "Un-mix: Rethinking image mixtures for unsupervised visual representation learning." AAAI 2022.
> 5. Zhong, Yuanyi, et al. "Is self-supervised learning more robust than supervised learning?." *arXiv preprint arXiv:2206.05259* (2022).
> 6. Huang, Weiran, Mingyang Yi, and Xuyang Zhao. "Towards the generalization of contrastive self-supervised learning." *arXiv preprint arXiv:2111.00743* (2021).
> 7. Ruan, Yangjun, Yann Dubois, and Chris J. Maddison. "Optimal representations for covariate shift." *arXiv preprint arXiv:2201.00057* (2021).

---

### Official Review · Reviewer_opzC · 2022-10-24

**Confidence:** 3
**Correctness:** 3
**Technical Novelty And Significance:** 2
**Empirical Novelty And Significance:** 2
**Recommendation:** 3

**Clarity, Quality, Novelty And Reproducibility:**

Sections 1 and 2 were generally clear. When defining the distance metric, "we define the Bhattacharyya distance" could be interpreted to mean that this paper came up with the distance metric to a reader unfamiliar with information geometry. Also, $\tau$ may not be the best notation for a trajectory - in classical differential geometry, $\tau$ is often used as a normalized time coordinate to parameterize a curve or manifold. $\gamma$ is a more standard notation for a curve.

In the section "Embedding a probabilistic model in lower-dimensions", I didn't understand the distinction between the InPCA method ("preserves global distances") and other dimensionality reduction methods ("preserves local distances"). Don't global distances get distorted when a small subspace is plotted? Or is the point that, like regular PCA, InPCA is based on reduction of the global set of distances?

Section 3 was very difficult to parse. There are many figures/subfigures, each with complicated explanations. Perhaps the text would be better suited to having fewer results, explained in more detail, with some moved to the appendix. In particular Figures 1 (c) and 3 were difficult to parse/interpret.

I believe Result 6 and Figure 4 were meant to be linked, but don't seem to be in the main text.

I believe the method is novel, but I am not well-versed in the fields of information geometry and contrastive learning, so I will defer to the other reviewers on that point. Many of the results are not novel. There is value of recapitulating known results with a new method, if that method can then be used to understand new phenomenology; however, in this work it's not clear if new results can be derived using this methodology.

**Strength And Weaknesses:**

Using the Bhattacharyya distance gives a nice way to generate normalized comparisons between different approaches. In addition, mapping sub-tasks to the union task gives a simple way to compare aspects of representation learning.

It is not clear to me that the time parameterization using the geodesic from the random initialization to the final solution is a good normalization in all situations. It may provide too much weight to early time learning dynamics. However, the fact that the authors derived a relatively simple normalization procedure is to their credit.

The claim of result 1 feels like an overreach. Dimensionality reduction techniques should not be conflated with dimensionality counting techniques. In addition, 33% of the variance explained is not a large quantity. While low dimensional dynamics exist in many ML settings (including perhaps this one), it is not clear what fractions of the dimensions are necessary to capture the relevant information. For example if much of the captured variance is early in the trajectories (when accuracy of the classifier is low), the low-dimensional picture may not capture the more interesting aspects of feature learning later in the trajectories.

Result 2 is perhaps less surprising, given results that the early layers of convolutional networks learn similar features in different tasks (e.g. https://link.springer.com/chapter/10.1007/978-3-319-10590-1_53). Result 3 is interesting in principle, but is very hard to see from the plot - again suggesting that the normalization may not be well-suited to answer certain questions.

Result 4 is also known. Result 5 is potentially interesting; however, the fact that the models arrive at similar points in information space is not surprising if both methods are known to produce solutions with similar accuracy. Result 6 is also potentially interesting, but I had trouble understanding the text and the figure, and their relationship.

**Summary Of The Paper:**

The authors use the Bhattacharyya distance to define notions like the distance between learning trajectories on classification tasks. They also define a simple notion of transfer learning, and use that to compute learning trajectory distances across different transfer tasks and

**Summary Of The Review:**

Overall, the use of information geometry + InPCA embedding is a very nice way to visualize learning trajectories with different setups. However, the overall results seem to mostly be known, and the presentation, particularly for the figures, can be hard to parse.

---

> ### Author Response · Authors · 2022-11-17
> **Response to Reviewer opzC (1/4)**
>
> Thank you for the feedback. We hope to convince you that the results in our manuscript are entirely novel, engage in a discussion and address your concerns.
>
> **>> Novelty, the overall results seem to mostly be known. Many of the results are not novel.**
>
> **We have been diligent in citing related work and in making quantitative and precise claims** **using sophisticated mathematical techniques. In the following rebuttal, we have provided a detailed reply to your comments.** **It is of course possible that we may have missed something.** **If you believe that our findings are not new, please** **point us to existing work that makes these claims precisely. We are happy to modify our claims. But we do not think it is reasonable to claim that these “results are mostly known” without providing any evidence.**
>
> **>>> however, in this work it's not clear if new results can be derived using this methodology.**
>
> The techniques developed in our paper are extremely general. These techniques can be used for any probabilistic model. So long as the likelihood p(y∣x) is well-defined, one can implement all the techniques developed in our paper. This includes problems beyond classification such as image segmentation (per-pixel likelihoods), tasks in natural language processing (likelihood of filling in a mask), and many others.
>
> **There is also a deeper contribution of this work that we did not discuss in the manuscript.** Most researchers would agree that it has been very difficult to use information geometry to understand mainstream questions in machine learning. So even if information geometry has developed a rich body of sophisticated ideas (e.g., Shun-ichi Amari’s book and many other seminal works), these ideas are locked away behind abstract mathematics and are difficult to wield computationally, especially for high-dimensional models such as deep networks.
>
> Our paper has developed a computational version of information geometry. This is by virtue of Equation 1 which is a finite-dimensional probability distribution (in contrast to the standard object in information geometry which is an infinite dimensional probability distribution defined over the entire domain of inputs x). This allows us to compute complex objects (embeddings of manifolds, geodesics, projections etc.). We bring sophisticated ideas and build computational tools and marshal them to the investigation of mainstream questions in modern deep learning. This might be a key technical contribution of our work that could have far-reaching impact. **To add, all this is entirely new. This paper presents a fresh perspective to understanding the representations learned by probabilistic models.**
>
> **>>> It is not clear to me that the time parameterization using the geodesic from the random initialization to the final solution is a good normalization in all situations. It may provide too much weight to early time learning dynamics.**
>
> There might be some misunderstanding here. Different models train at very different speeds, in particular at the beginning of training models move rapidly after each mini-batch update in the space of probability distributions (e.g., as measured by the Bhattacharyya distance). We can only record the trajectory at specific checkpoints (e.g., after each epoch or at best after each mini-batch). Our time re-parameterization technique allows us to interpolate two successive checkpoints using the geodesic that connects them. We discretize “progress” t_w into 100 equidistant intervals and interpolate the entire trajectory at these 100 points by calculating the appropriate points on the geodesic between two successive checkpoints; this is described in the narrative after Equation 4. For initial parts of the trajectory, there are fewer checkpoints per unit progress because the network trains quickly. But the time re-parameterization still allows us to discretize the entire trajectory evenly. **Therefore the time re-parameterization technique actually provides equal weights to early and late times of the training process.**

---

> > ### Comment · Reviewer_opzC · 2022-11-18
> > **Thanks for the detailed responses**
> >
> > I will reply to each set of comments individually.
> >
> > With regards to the normalization: I think this point:
> >
> > "in particular at the beginning of training models move rapidly after each mini-batch update in the space of probability distributions (e.g., as measured by the Bhattacharyya distance)"
> >
> > is the one that makes me wonder about the use of the geodesics, especially in the context of using the metric for things like PCA. I do appreciate the clarification, and want to make it clear that I think any method will have tradeoffs; I do believe that the choice the authors made is a reasonable one. Nevertheless I thank the authors for the detailed discussion/clarification on this point.

---

> > > ### Author Response · Authors · 2022-11-19
> > > **Thank you for the response**
> > >
> > > Thank you for your quick response. We appreciate your time and open-mindedness. We believe we have addressed most of your concerns from the original review, and have also responded to your latest comments. Since you seem to be broadly in agreement about the merits of our work after our response, we would like to encourage you to reconsider your score.

---

> ### Author Response · Authors · 2022-11-17
> **Response to Reviewer opzC (2/4)**
>
> **>>> The claim of result 1 feels like an overreach. Dimensionality reduction techniques should not be conflated with dimensionality counting techniques**
>
> **First note that, in our entire paper, visualizations using InPCA are only used to provide a qualitative interpretation of the results. All claims in the paper are based on exact calculations, e.g,. distances between trajectories in Plots 1g, 3b, 4b, use the Bhattacharyya distance between models/trajectories---and not the distance in the InPCA co-ordinates.**
>
> Next let us appreciate some numbers. The probabilistic models in Fig 1a lie in a 50 million dimensional space (50,000 validation images of Imagenet and 1000 classes). In Fig 1b, top 3 dimensions capture about 33% of the variance of points. In these 50 million dimensions, trajectories of these models (with ~23 million weights for a ResNet-50) trained from different initializations and on different datasets could be very complicated. **But they seem to lie on a manifold that effectively has ~0.003% of the dimensionality. This also means that the effective dimensionality of models from different initialization trained on one task is this small. You might agree that this is totally unexpected. We do not know any other theoretical or empirical result that predicts this phenomenon.**
>
> We completely agree with you that dimensionality reduction and dimensionality counting are not the same thing. We have been careful to say in our manuscript that the manifold is “effectively low-dimensional”, our definition for this being the explained variance. Result 1 is saying that this effective dimensionality is surprisingly small.
>
> We have also made the following change to the manuscript.
>
> **Before:** In the original manuscript, we were using
> $$  \text{Explained variance}(k) = \frac{\sum_{i=1}^k |\lambda_i|}{\sum_{i=1}^n |\lambda_i|} $$
>
> where $\lambda_i$ are the eigenvalues of the centered distance matrix W, defined in Equation 9. We used the absolute values of the eigenvalues to work around the fact that some of them are negative in this Minkowski space. This expression for explained variance is inspired by the expression for standard principal component analysis. For standard PCA, such an expression ($\lambda_i≥0$) measures how well the projections into lower dimensions can be used to recover the original points. But this is not appropriate because the matrix D in Equation 9 is not the matrix of correlations of points (like PCA), it is instead the Bhattacharyya distance between models.
>
> **Now:** In the past couple of months, we realized that the more appropriate metric to use for our work is instead “the fraction of pairwise distances that are preserved from the original space of probabilistic models in the embedded coordinates”. This can be calculated to be (see Appendix D for a derivation)
> $$ \text{Explained stress}(k) = 1 - \sqrt{\frac{\sum_{i=k+1}^m \lambda_i^2}{\sum_{i=1}^m \lambda_i^2}} $$
> This quantity is derived from the objective function of multi-dimensional scaling (which is often called “stress”). We can calculate the explained stress without calculating all the eigenvalues using the identity in Equation 12.
>
> We show the explained stress in Fig 1b for the data in Fig 1a; the top 3 dimensions explain ~80% of the stress. In other words, on average, the distance between any two models as measured by the top 3 dimensions (k=3 on the left hand side of Equation 10) is ~80% accurate as compared to the Bhattacharyya distance in the original space (right hand-side of Equation 10).
>
> **To emphasize, we are not changing from “explained variance” to “explained stress” to boost our number from 33% to 80%. The original choice was incorrect---it is valid for PCA but not for InPCA/MDS.**
>
> **>>> if much of the the captured variance is early in the trajectories, the low-dimensional picture may not capture the more interesting aspects of feature learning later in the trajectories.**
>
> **The three-dimensional visualization is provided solely for the purpose of interpreting the results of these experiments qualitatively. None of the conclusions in our paper is drawn from the visualization.**
>
> If our goal is to investigate the properties of the training trajectories in different regimes (e.g., at the beginning, in the middle, and at the end), we can calculate the embeddings, and estimate the distances between trajectories in Equation 5 using only that part of the data. This can potentially lead to very interesting findings. In the present paper, we are interested in understanding the space of learnable tasks and that is why we used the entire trajectories. It is a testament to the generality and rigor of the techniques developed in this paper that they can be used to investigate different questions pertaining to learning dynamics. We hope other researchers will use these methods to investigate their hypotheses. And therefore we commit to making the code, trained models and interactive visualizations public.

---

> > ### Comment · Reviewer_opzC · 2022-11-18
> > **Dimensionality reduction**
> >
> > Two examples showing low-dimensionality of learning dynamics:
> >
> > https://arxiv.org/abs/1812.04754
> >
> > https://proceedings.neurips.cc/paper/2018/hash/7a576629fef88f3e636afd33b09e8289-Abstract.html
> >
> > The second reference in particular shows that even in the null model of a random walk, there can be concentration of the variation in a small number of dimensions. Though these methods did not show the dimensionality in this comparative, information theoretic way, the basic idea that a small number of dimensions can explain a large fraction of the variation in learning trajectories is well-studied.

---

> > > ### Author Response · Authors · 2022-11-19
> > > **Re: Dimensionality reduction**
> > >
> > > **>>> Though these methods did not show the dimensionality in this comparative, information theoretic way, the basic idea that a small number of dimensions can explain a large fraction of the variation in learning trajectories is well-studied.**
> > >
> > > Thanks. We have cited both these papers in Related Work now. Our work differentiates itself from both https://arxiv.org/pdf/1812.04754.pdf and  https://proceedings.neurips.cc/paper/2018/file/7a576629fef88f3e636afd33b09e8289-Paper.pdf in the following ways:
> > >
> > > 1. The mentioned works discuss the low-dimensionality of trained trajectories in the weight-space for a model trained on one task while **our work shows effective low-dimensionality in the prediction space**. Even if these papers are also about the learning dynamics of neural nets, our results do not follow from these results. Further, the techniques that we use to analyze trajectories in the prediction space are totally different.
> > > 2. The two works make statements about the learning dynamics of one supervised learning model. In our work, we claim that models trained on **different tasks and using different methods** are also effectively low-dimensional in prediction space. **This is a totally different problem** and does not follow from these existing results.

---

> ### Author Response · Authors · 2022-11-17
> **Response to Reviewer opzC (3/4)**
>
> **>>> Result 2 is perhaps less surprising, given results that the early layers of convolutional networks learn similar features in different tasks  (e.g.** [**https://link.springer.com/chapter/10.1007/978-3-319-10590-1_53**](https://link.springer.com/chapter/10.1007/978-3-319-10590-1_53))
>
> We do not understand your argument. The paper that you have linked to shows that low-level filters learned on different datasets are qualitatively similar; this has been very widely studied since 2014. This does not mean that models trained on different tasks have similar trajectories in the space of probability distributions. Result 2 is not just a statement about filters, it is a statement about the entire 50 million-dimensional prediction vector (50,000 images and 1000 output classes). It is saying that training on the task “Dogs” makes progress on seemingly dissimilar tasks like “Instruments” (which contains vehicles, devices and clothing). In fact, the Dogs model predicts on the entire Imagenet as well as a model trained on the _entire_ Imagenet for about 63.38% of the progress. If the task is more diverse (say 333 random classes), then the progress on the entire Imagenet is as large as 92%.
>
> Of course the reason that models trained on one task predict well on another task is because there is some shared structure (as we say in the Introduction). And because of this shared structure, there has also been some evidence that filters of models trained on different datasets look similar. **But this previously found qualitative similarity between some filters of some tasks does not imply our result. Our result is a precise quantitative statement about the training trajectory and the entire representation.**
>
> **>>> Result 3 is interesting in principle, but is very hard to see from the plot**
>
> Let us explain. Fig 1f gives us a visual interpretation of the result. **The precise way to understand Result 3 is using the quantitative analysis reported in Fig 1g (and FigA1.2) like we have explained in the caption.** The X-axis of the plot is “progress” (Equation 4). For multiple models (5 random seeds) trained on two tasks (say Conveyance and Dogs), we calculated the mean (across the random seeds) of the interpolated trajectories at different progress. At each progress, we then plotted the distance between the mean model trained on Conveyance (say task 1) and Dogs (say task 2) divided by the average tube radii (which is the maximum of the distance of the model of one seed from the mean):
>
> $$y(t_w) = \frac{2 \text{dB}(\tau_{\text{mean}}^{1 \to U}, \tau_{\text{mean}}^{2 \to U})}{\max_a[\text{dB}(\tau_a^{1 \to U}, \tau_{\text{mean}}^{1 \to U})] + \max_a[\text{dB}(\tau_a^{2 \to U}, \tau_{\text{mean}}^{2 \to U})]} $$
>
> The is a measure of how far away the trajectories of these two models are. If the y(tw​)<1, then the “tubes” corresponding to models trained on tasks 1 and task 2 intersect.
>
> **>>> Result 4 is also known**
>
> Can you point to existing work that makes this claim precisely and quantitatively? We are happy to cite and modify this claim. We have already discussed existing work that has noticed similar phenomena in the narrative for Result 4.
>
> **>>> Result 5 is potentially interesting; however, the fact that the models arrive at similar points in information space is not surprising if both methods are known to produce solutions with similar accuracy**
>
> The episodic meta-learning objective differs from the objective used in supervised learning. Supervised learning drives the training error to zero since it minimizes the cross-entropy loss over all the classes. However, episodic meta-learning methods optimize a loss that considers all k-way classification tasks (k is typically smaller than the total number of classes in the dataset C). **Since the two objectives are different, it is not obvious that both methods will have high training accuracies at the end.** It is true that a model that minimizes the cross-entropy loss also minimizes the episodic learning objective. But the reverse is not necessarily true.
>
> These experiments actually show a more important point. It is not just that the final representations of supervised learning and episodic meta-learning are similar. The **entire trajectories are similar if we use a large way (trajectory for 5-way is very close to that of supervised learning). This is surprising and certainly a new observation.**

---

> > ### Comment · Reviewer_opzC · 2022-11-18
> > **Training on different tasks**
> >
> > The point I was trying to make with the work on the early convolutional layers is that, in your setup, there is transfer across tasks if the features derived from the forward pass of the network on task 1 (before the final layer) are useful for learning on task 2 (linear regression on top of those features). In the work I referenced, the similarity of the learned filters was due to their usefulness across tasks - which suggests that transfer could be useful.
> >
> > Perhaps a better link would have been to transfer or few shot learning literature, where this re-use of feature information is in fact the effect which gets exploited.
> >
> > Regarding result 4: there is a vein of work analyzing the [neural tangent kernel](https://proceedings.neurips.cc/paper/2018/hash/5a4be1fa34e62bb8a6ec6b91d2462f5a-Abstract.html) and showing that it can be used to approximate the results of finetuning:
> >
> > https://proceedings.neurips.cc/paper/2020/file/405075699f065e43581f27d67bb68478-Paper.pdf
> > https://proceedings.mlr.press/v162/wei22a.html
> > https://arxiv.org/abs/2210.05643
> >
> > (The first paper does not study fine-tuning/transfer learning explicitly, but still shows that typically during training the features eventually "freeze".) There may be similar results in the more general transfer learning/finetuning literature but I am not as familiar with those.
> >
> > Regarding result 5: since the two objectives are highly correlated, it is still not clear to me why the endpoint similarity is surprising. The result states `even if the two traverse very different trajectories during training.`; if the similarity of the trajectories is true and a key part of the result, it should be stated more explicitly (I did see the discussion of this point as ways increase at the end of that section).

---

> > > ### Author Response · Authors · 2022-11-19
> > > **Re: Training on different tasks (1/2)**
> > >
> > > **>>>  In the work I referenced, the similarity of the learned filters was due to their usefulness across tasks - which suggests that transfer could be useful.**
> > >
> > > Exactly, the emergence of qualitatively similar filters in models trained on different tasks is pointing to some shared structure in the space of learnable tasks. There have been numerous such results since 2014 (when the paper that you mentioned was published), and as we say in the introduction, this is our motivation as well. But we do not exactly know what this shared structure is, or how to measure it quantitatively. Our work provides a crisp quantitative way to characterize the structure in the space of learnable tasks. It is not the complete story of course, since we do not know “why” tasks seem similar.
> > >
> > > **>>> Perhaps a better link would have been to transfer or few shot learning literature**
> > >
> > > Indeed, as we discuss quite elaborately in the Related Work section, lots of work tries to exploit the shared structure between tasks, and formalize what this shared structure could be using ideas from learning theory/information theory, etc. Such ideas have given rise to entire fields of research, e.g., transfer, multi-task, meta, few-shot, self-supervised learning etc. **The progress in these fields has been quite impressive and that is why our goal in this paper is to provide a quantitative way to understand the shared structure** **between tasks. Our findings are entirely new; they reveal why algorithms in these above fields work.**
> > >
> > > **We wanted to convey our point of view of the utility of an “intuitive” result.** Researchers working in the sub-fields mentioned above (we are too) have various intuitions as to why their algorithms work. These intuitions are exactly what we wanted to test. But just because something may be intuitive (e.g., episodic learning being “similar” to supervised learning), it does not mean that there is a scientifically precise understanding of the phenomenon. Intuition is a double-edged sword. For example, intuition suggests that models with millions of weights fitted on a non-convex energy landscape on seemingly dissimilar tasks are unlikely to learn similar representations. But since transfer learning is so effective, it is also intuitive that the representations are similar. And this is why---to ground such intuition and make it systematic---we need precise quantitative techniques to study such phenomena. Our paper builds very general techniques to study probabilistic models to do so.
> > >
> > > **>> Regarding result 4: there is a vein of work analyzing the neural tangent kernels and showing that it can be used to approximate the results of finetuning**
> > >
> > > Thanks. We have cited these works in the latest version of the manuscript.
> > >
> > > **There is merit to understanding pre-training/fine-tuning from different perspectives.** A number of papers have done this; we have also cited some in the narrative in Result 4. Wei et al. use random matrix theory (https://proceedings.mlr.press/v162/wei22a.html) while Malladi et al. (https://arxiv.org/abs/2210.05643) consider the NTK model.
> > >
> > > In Result 4 (and Figure 3), we look at the trajectories of representations when fine-tuned from different points on the supervised learning trajectory. **Wei et al. and Malladi et al. do not make any claims on the trajectories of fine-tuning---even in weight space.** Also, calculations using theoretical models is rather different from showing quantitative evidence on a real network with all bells and whistles. The technical tools developed in our work to analyze trajectories are very different and consider an information geometric perspective. **We can use our tools to study the trajectory of any representation learning algorithm for any probabilistic model.**
> > >
> > > **>>> The first paper does not study fine-tuning/transfer learning explicitly, but still shows that typically during training the features eventually "freeze".** ([https://proceedings.neurips.cc/paper/2020/file/405075699f065e43581f27d67bb68478-Paper.pdf)](https://proceedings.neurips.cc/paper/2020/file/405075699f065e43581f27d67bb68478-Paper.pdf)
> > >
> > > This is very different from our work. It shows that the features eventually freeze when trained **on a single task---there is no fine-tuning**. **This result is also reflected in our paper: end points of trajectories of 5 random seeds in Fig 1a are quite similar.** Result 4 and Figure 3 discuss the scenario where we first pre-train the model (for different number of epochs) and then fine-tune on a specific task. We find that the change in the representations depends on the distance of the the pretrained model to P*. So the representation does change a lot depending upon where fine-tuning is begun from. As we discuss in the paper, this also has a powerful empirical implication: depending upon the target task, it may be useful to not train on the source-task completely and instead fine-tune from some intermediate point on the trajectory.

---

> > > ### Author Response · Authors · 2022-11-19
> > > **Re: Training on different tasks (2/2)**
> > >
> > > **>>> Regarding result 5: since the two objectives are highly correlated, it is still not clear to me why the endpoint similarity is surprising**
> > >
> > > **Can you please explain what you precisely mean by “two objectives are correlated”?** The objectives of episodic learning and supervised learning are clearly different from each other. We do not know of any existing proof that these two techniques reach the same solution. Indeed, if it were so intuitive that they reach the same solution, then episodic learning would not be a field of research!
> > >
> > > **>>> if the similarity of the trajectories is true and a key part of the result, it should be stated more explicitly**
> > >
> > > Thanks, we have made this more explicit now in Result 4. We calculated the distance between the trajectories of supervised learning and episodic learning using a different number of ways. The results are plotted in Appendix G of the updated paper.
> > >
> > > The maximum distance at the same progress (normalized by the tube radii) between models trained with 5-way episodic learning and those trained with supervised learning is about 12x smaller than the maximum distance at the same progress between models trained with 2-way episodic learning and those trained with supervised learning (left most panel in Fig A3). **This suggests that the 5-way episodic learning trajectory is really very similar to that supervised learning. To add, we had noted in the original manuscript that the 2-way trajectory is 40x longer in Riemann length than that of supervised learning.**

---

> ### Author Response · Authors · 2022-11-17
> **Response to Reviewer opzC (4/4)**
>
> **>>> Result 6 is also potentially interesting, but I had trouble understanding the text and the figure, and their relationship.**
>
> We again use Fig 4a and 4b in conjunction to understand Result 6. We have color-matched the lines in Fig 4b with those in Fig 4a now. The black curve is the trajectory of supervised learning on the entire CIFAR-10; red is the trajectory of SimCLR trained on the entire CIFAR-10. Fig 4b compares the distances of trajectories in Fig 4a from the red one “contrastive”; this is why there is no red trajectory in Fig 4b.
>
> * The first thing to note here is that the black and red trajectories are quite close to each other; the black line in Fig 4b is only about 20 times far away from red as compared to their corresponding tube radii.
> * Next observe that the trajectory of SimCLR on Task 1 (light blue), SimCLR on Task 2 (green) and SimCLR on Task 3 (yellow) are very similar to each other; this is seen in both Fig 4a and in Fib 4b.
> * Third, they are closer to SimCLR on all of CIFAR-10 than any supervised learning trajectories (this is seen in Fig 4b because their curves are below everyone else). Thus, contrastive learning on datasets with different classes learns similar representations.
> * The learned representation of two-class SimCLR models is similar to the one obtained using data from all classes (red) (in this experiment this occurs up to about t_w = 0.4 progress) but they do not go all the way to the truth (i.e., the end point of black line). This shows the benefit of having data from many classes during contrastive learning.
> * In the updated manuscript, we have also added a new experiment in Appendix F that compares Barlow Twins (contrastive learning) and Fixmatch (semi-supervised learning). The trajectories of these methods are similar to those of SimCLR.
>
> We have also shown distances computed with respect to other trajectories in Fig A1 which can be used to further investigate these claims.
>
> **>>> \gamma is a more standard notation for a curve**
>
> Please allow us this editorial choice.
>
> **>>> “we define the Bhattacharyya distance" could be interpreted to mean that this paper came up with the distance metric**
>
> We said “we define” to avoid confusion for the informed reader: the Bhattacharyya distance is usually not normalized by the number of samples N. Normalizing it by N helps us interpret the numerical values of distances more easily. In any case, we have cited a reference for the Bhattacharyya distance now.
>
> **>>> I didn't understand the distinction between the InPCA method. Don't global distances get distorted when a small subspace is plotted?**
>
> Yes. We are using a dimensionality reduction technique (InPCA) which has the specific property that if one uses all the eigenvalues (i.e., the dimensionality of the embedding space is equal to the number of probabilistic models being embedded) then the pairwise Bhattarcharyya distances between points are preserved exactly. When we visualize the top three dimensions of the embedding, we only see a partial picture of the manifold. Pairwise distances between points are distorted. We have modified the narrative now to avoid the confusion. Note that the visualizations are only for providing intuition and interpretability; all claims in the paper are made quantitatively on the basis of the Bhattacharyya distances between trajectories.
>
> **>>> Section 3 was very difficult to parse.**
>
> We appreciate this comment. This paper develops a lot of technical machinery and discusses results that touch many different sub-areas of machine learning. This is admittedly a lot of material to put into a 9-page manuscript. We have tried to write precisely although the length restrictions forced us to be extremely terse. We have tried to strike the balance in such a way that even if it takes some extra time to read the manuscript for all readers, the manuscript should be precise enough that most readers should not be able to misunderstand and mis-interpret our results.
>
> In the updated version of the manuscript, we have expanded the Appendix significantly to provide more details on the calculations, and some additional comparisons and results.

---

> > ### Comment · Reviewer_opzC · 2022-11-18
> > **Thanks for the comments**
> >
> > Thank you for the clarifications on Result 6. I think I understand it better now, but overall the readability is still low. I do appreciate the point that the length makes it hard to develop technical material; perhaps that means that less material should be presented in more detail.
> >
> > Thanks also for adding the reference for Bhattacharyya distance; as someone who is not an info theory practitioner it helps quite a bit!
> >
> > I still would rather the curve not be defined as $\tau$, but my approval does not hinge on this point.

---

> > > ### Author Response · Authors · 2022-11-21
> > > **Re: Thanks for the comments**
> > >
> > > **>> I think I understand it better now, but overall the readability is still low. I do appreciate the point that the length makes it hard to develop technical material; perhaps that means that less material should be presented in more detail.**
> > >
> > > We commit to moving Result 4 to the Appendix. Using the additional half a page or so of space, we will expand the narrative for Result 1 and Result 6. Since we cannot update the PDF after Nov 18th, we will not be able to show you the updates but we commit to modifying the camera-ready version of the paper.
> > >
> > > Specifically, we will add the content in the rebuttal on “how to interpret Fig 1g, 3b etc.” (this was in response to your question) and how to calculate the trajectories in Fig 1a, 1f etc. (this was in repose to wHo9’s question titled “I don't quite understand the "n" or equation (9) and the trajectories plotted in figure 1").

---

### Official Review · Reviewer_wHo9 · 2022-10-24

**Confidence:** 3
**Correctness:** 3
**Technical Novelty And Significance:** 3
**Empirical Novelty And Significance:** 3
**Recommendation:** 6

**Clarity, Quality, Novelty And Reproducibility:**

The clarity can be improved - see comments above. Methods and tools seem novel. Code is not submitted and code release is not discussed which is a weak point.

**Strength And Weaknesses:**

Strengths:

-- The paper proposes a set of tools that can be used to analyze neural network training, trajectories and similarities in general, even for models on different tasks.

-- The paper makes several interesting observations on the representations learned.

Weaknesses:

-- The metric Bhattacharyya distance relies only on the output y, so even if the intermediate representation is very different it still calculates two models as very close. This may not be ideal, especially when two models agree on the output but not representations. The later tools all rely on this distance so this affects all metrics and observations in this paper.

-- Given the point above, when comparing models trained on the same task (e.g., supervised learning and meta-learning), it may not come as a surprise that their eventual similarity is high, since both of them will get high accuracy in the end.

-- I feel the presentation and clarity can be greatly improved, especially for audience without a prior background in this topic.

-- The paper's introduction does not cite any prior work which is weird. I also feel the six listed findings do not have to repeat in introduction and abstract, maybe only one place is enough.

Other comments:
-- Remark 1: "Note that if we were to train the classifier w2 (with backbone w1 fixed) using samples from the other task under the constraint that rows of w2 have unit l2 norm, then we would obtain the imprinted weights as our solution." I get that imprinting is a way to substitute linear probing, but Is there a mathematical proof for this statement?

-- Equation (2's) "*" step is still not that clear to me. Maybe it could be made more clear by having an additional intermediate step.

-- "where Lij = δij − 1/n is the centered version of D." The centered version of D seems to be W.

-- The last equation in Section 2, can you make it more clear why it indicates isometry?

-- The axes and ticks in figures can be made thinner and more transparent to improve the visualization and highlight the trajectories.

-- I don't quite understand the "n" for equation (9) and the trajectories plotted in figure 1. Given that each epoch produces a standalone model, for an intermediate model, where to select the other n-1 models to construct a joint model embedding?

To summarize, the paper proposes a set of useful tools to analyze and compare different models, and present interesting findings with them from the perspective of task relationships. But the paper also has many points to improve on. Therefore I will recommend "marginally above".

**Summary Of The Paper:**

The paper develops a set of tools to measure model distances, and visualize model embeddings/training trajectories, and thus make measurements on different tasks' relationship. Techniques used includes Bhattacharyya distance, linear layer imprinting, InPCA and others. The paper then uses these tools to make several interesting observations on self-supervised learning, meta learning, supervised learning and fine-tuning and find the representations are quite shared.

**Summary Of The Review:**

To summarize, the paper proposes a set of useful tools to analyze and compare different models, and present interesting findings with them from the perspective of task relationships. But the paper also has many points to improve on. Therefore I will recommend "marginally above".

---

> ### Author Response · Authors · 2022-11-17
> **Response to Reviewer wHo9 (1/2)**
>
> We are glad that you find these results interesting and appreciate the points raised in the review. We have modified the manuscript at various places (in green) to address your comments.
>
> **>>> Bhattacharyya distance relies only on the output y, so even if the intermediate representation is very different it still calculates two models as very close**
>
> This is actually a very desirable property. If two different representations give the same probabilistic model there is nothing to distinguish them from each other. Consider a representation and a rotated version of the same representation. Even if the Euclidean distance between the two is large, they are identical for the purposes of the classification task. By considering only the output y, we are able to capture such symmetries and assign the distance between the two representations to be small.
>
> **>>> when comparing models trained on the same task (e.g., supervised learning and meta-learning), it may not come as a surprise that their eventual similarity is high, since both of them will get high accuracy in the end.**
>
> The episodic meta-learning objective differs from the objective used in supervised learning. Supervised learning drives the training error to zero since it minimizes the cross-entropy loss over all the classes. However, episodic meta-learning methods optimize a loss that considers all k-way classification tasks (k is typically smaller than the total number of classes in the dataset C). **Since the two objectives are different, it is not obvious that both methods will have high training accuracies at the end.** It is true that a model that minimizes the cross-entropy loss also minimizes the episodic learning objective. But the reverse is not necessarily true.
>
> These experiments actually show a more important point. It is not just that the final representations of supervised learning and episodic meta-learning are similar. The **entire trajectories are similar if we use a large way (trajectory for 5-way is very close to that of supervised learning). This is surprising and certainly a new observation.**
>
> **>>> I get that imprinting is a way to substitute linear probing, but Is there a mathematical proof for this statement?**
>
> We have added the following proof to the manuscript in Appendix E. Let the set of all samples in class $c$ be {$x_i^c$}$._{i=1}^{n_c}$. The log-probability that these samples belong to class $c$ is
>
> $$\sum_{i=1}^{n_c} \log p(y=c \mid x_i^c) = \sum_{i=1}^{n_c}  w_c \cdot \varphi(x_i^c) = w_c . (\sum_{i=1}^{n_c} \varphi(x_i^c))$$
>
> where $w_c$ are the weights for class $c$, and $\varphi(x_i)$ are the features for sample $x_i$.
>
> We would like to find a $w_c$ --- for each class c --- such that the above log-probability is maximized subject to the constraint that $||w_c||=1$. If we don’t have this constraint, we can maximize the above expression by sending $w_c \rightarrow \infty$. Note that $w=x/||x||$ maximizes  $w \cdot x$, subject to the constraint that $||w||=1$. Hence, the optimal value for $w_c$ is $\sum_{i} \varphi(x_i^c)/||\sum_{i} \varphi(x_i^c)||$.
>
> **>>> Equation (2's) "*" step is still not that clear to me.**
>
> We have added Appendix C to the manuscript which explains step (*). The step is similar to how the entropy of the joint distribution of two independent random variables is the sum of their individual entropies.

---

> ### Author Response · Authors · 2022-11-17
> **Response to Reviewer wHo9 (2/2)**
>
> **>>> I don't quite understand the "n" or equation (9) and the trajectories plotted in figure 1**
>
> “n” is the number of models used to compute the embedding. For example, in Figure 1(a), we train on 6 tasks, 5 random seeds for each task, and record 61 model checkpoints over the course of training (for each of the 6x5=30 training runs). Hence, there are a total of n=6x5x61=1830 Imagenet models in Figure 1(a).
>
> Pairwise distances are denoted by a matrix D of size 1830x1830. To compute the embedding, we first center the distance matrix D using Eq. 9 and then perform an eigen-decomposition. Equation 9 is a classical dimensionality-reduction method called multi-dimensional scaling (MDS). But as we discussed, the way we are using it is novel because we do not throw away the negative eigenvalues. This allows the embedding to preserve pairwise distances between points. This is if all eigenvectors are used, otherwise there are some distortions in pairwise distances which we quantify using a quantity called “explained stress” (see Appendix D). Fig 1a plots the 1830 models in a 3-dimensional sub-space of this embedding. If we compute distances in these 3-dimensions, it is a surprisingly good approximation to the original distance matrix, which suggests that these models lie on a low-dimensional manifold.
>
> For each trajectory (one particular task and one particular random seed), we use the procedure discussed in Equations 3-4 to reindex all its checkpoints. We interpolate between each of the 61 checkpoints using the geodesic in Equation 3. Each training trajectory can now be evaluated at any progress t_w in [0, 1] (defined in Equation 4). Given 5 random seeds of each task, we can calculate their “average trajectory” by averaging the output probabilities at a fixed value of t_w for all the interpolated trajectories; 100 different values of t_w spread uniformly between [0,1] are chosen. These 100 points along the average trajectory of each task are also embedded together with the 1830 checkpoints (i.e., n=1830 + 5x100 = 2330). And that is how we can plot the “trajectory“ in Fig. 1a.
>
> We would like to emphasize that the above steps are highly non-trivial. Both in terms of the computational complexity of calculating the embedding and interpolating such a large number of models but also in terms of the technical complexity of implementing ideas from information geometry such as the InPCA embedding, geodesics in these high-dimensional spaces and projections onto these geodesics. **The techniques developed in our paper to study the space of tasks are very general and very fresh.**
>
> **These techniques that we have developed are very different from anything used to study probabilistic models today. These are precise techniques and that is why we can reveal some remarkable phenomena. Some are startling, e.g., representations learned on different tasks are low-dimensional. The content of our paper is very novel and could lead to a foundational understanding of representation learning.**
>
> **>>> where $L_{ij} = \delta_{ij} − 1/n$ is the centered version of D.“**
>
> Thank you, we have fixed this now.
>
> **>>> paper's introduction does not cite any prior work**
>
> We have cited relevant prior work (~70 references) in the Related Work section and also point to this section in the introduction (we made it more prominent now). We would like to discuss related work in the context of the results. We feel that it is difficult for an uninformed reader to understand the purpose of the references if they come upfront before the manuscript develops any of its technical ideas. We hope you agree with this editorial choice of ours.
>
> **>>>  Code is not submitted and code release is not discussed which is a weak point.**
>
> We commit to releasing all code, data (model checkpoints, model embeddings), and interactive visualizations of the embeddings with the final version of the paper.

---

### Author Response · Authors · 2022-11-17
**Response to all reviewers**

We thank the reviewers for their feedback. We are glad that the reviewers find the developed tools to be useful and novel [opzC, wHo9] and the findings to be interesting [wHo9, hj5s]. We are also glad that Reviewer hj5s finds the paper to be well organized. We have improved clarity by significantly expanding the appendix section. Changes to the paper are highlighted in green.

We have addressed the concerns of the reviewers as individual responses below.

---

### Decision · Program_Chairs · 2023-01-20

**Decision:**

Reject

**Justification For Why Not Higher Score:**

I, the AC, actually liked this paper, and thought it was neat. However, none of the reviewers were really willing to argue for acceptance, and it is the case that incorporating the reviewer comments would make a stronger paper.

**Justification For Why Not Lower Score:**

N/A

**Metareview: Summary, Strengths And Weaknesses:**

This paper introduces a means for visualizing representations learned in many different settings (supervised, self-supervised, meta-learned, etc). This leads to deriving insights across these different learning tasks, and the degrees to which they do (or do not) learn similar or related representations. It's an interesting paper which stands out as something slightly "different", which is commendable.

Ultimately, the reviewers felt the paper was a bit borderline in its current form, and none were able to argue strongly for acceptance. I do think that the comments from the reviewers (along with the responses from the authors) help identify some of the major questions one might have when reading this work, and I believe incorporating them into a resubmission would make for a stronger paper overall.